

# Investigating the liquid water path over the tropical Atlantic with synergistic airborne measurements

Marek Jacob[1], Felix Ament[2], Manuel Gutleben[3], Heike Konow[2], Mario Mech[1], Martin Wirth[3], and Susanne Crewell[1]

[1]Institute for Geophysics and Meteorology, University of Cologne, Albertus-Magnus-Platz, 50923 Cologne, Germany
[2]Meteorological Institute, University of Hamburg, Bundesstrasse 55, 20146 Hamburg, Germany
[3]German Aerospace Center, Münchener Str. 20, 82234 Oberpfaffenhofen-Wessling, Germany

**Correspondence:** Marek Jacob (marek.jacob@uni-koeln.de)

**Abstract.** Clouds are a strongly variable component of the climate system and several studies have identified especially marine low level clouds to play a critical role for the climate. Liquid water path (LWP) is an important quantity to characterize clouds. Passive microwave satellite sensors provide the most direct estimate on global scale, but suffer from high uncertainties due to large footprints and the superposition of cloud and precipitation signals. Here, we use high spatial resolution airborne

microwave radiometer (MWR) measurements together with cloud radar and lidar observations to better understand LWP of warm clouds over the tropical North Atlantic. The nadir measurements were taken by the German High Altitude and Long range research aircraft (HALO) in December 2013 (dry season) and August 2016 (wet season) during two Next generation Advanced Remote sensing for VALidation campaigns (NARVAL).

Microwave retrievals of integrated water vapor (IWV), LWP and rain water path (RWP) are developed using artificial neural

network techniques and a unique database based on cloud-resolving model simulations with 1.25 km grid spacing. The IWV and LWP retrievals share the same eight MWR frequency channels as their sole input. The comparison of retrieved IWV with coincident dropsondes and water vapor lidar measurements shows root-mean-square deviations below $1.4\,\mathrm{kg\,m^{-2}}$ over the range from $20\,\mathrm{kg\,m^{-2}}$ to $60\,\mathrm{kg\,m^{-2}}$. This comparison raises the confidence in LWP retrievals which can only be assessed theoretically. The theoretical analysis shows the dependency of the uncertainty on LWP itself as the error is below $20\,\mathrm{g\,m^{-2}}$ for LWP below

$100\,\mathrm{g\,m^{-2}}$ and below $20\,\%$ above. The identification of clear sky scenes by ancillary measurements, here backscatter lidar, is crucial for thin clouds (LWP $< 12\,\mathrm{g\,m^{-2}}$) as the microwave retrieved LWP uncertainty is higher than $100\,\%$. The RWP retrieval combines active and passive microwave observations and is able to detect drizzle and light precipitation.

The analysis of both campaigns reveals that clouds were more frequent in the dry than in the wet season and their LWP and RWP were higher, but microwave scattering of ice was observed more frequently in the wet season ($1.6\,\%$ vs. $0.5\,\%$ of

the time). As to be expected, the observed IWV clearly shows that the wet season ($\overline{\mathrm{IWV}} = 41\,\mathrm{kg\,m^{-2}}$) is more humid than the dry season ($\overline{\mathrm{IWV}} = 28\,\mathrm{kg\,m^{-2}}$). The results reveal that the observed frequency distributions of IWV are strongly affected by the choice of the flight pattern. Therefore, the airborne observations need to be used carefully to mediate between long-term ground-based and spaceborne measurements to draw statistically sound conclusions.



## 1 Introduction

Clouds and precipitation are a fundamental part of the Earth's climate system and significantly contribute to the water and energy cycle. However, the great variability of clouds, the complex interaction of small-scale processes involved, and their coupling to atmospheric circulation make them a major source of uncertainty in numerical climate and weather models (e.g., Bony et al. (2015), Boucher et al. (2013)). Especially, shallow marine clouds are attributed to contribute largely to intermodel spread of climate models (Sherwood et al., 2014). These clouds are particularly difficult to assess from spaceborne sensors due to their small size, with about 70 % appearing in sizes of less than 2 km over the tropical North Atlantic (Schnitt et al., 2017). The accurate observation of thin liquid clouds is an ongoing and important challenge as they cover more than a quarter of the globe and are an important contribution to Earth's energy balance (Turner et al., 2007).

Liquid water content (LWC) is the key parameter to describe clouds in atmospheric models. Due to the even higher difficulty in observing LWC profiles (Crewell et al., 2009), we focus on the liquid water path (LWP). It describes the total mass of all liquid water droplets in an atmospheric column above a unit surface area. However, care has to be taken whether LWP only denotes the contribution by cloud droplets, later on called CLWP, or whether it also includes the contribution by liquid precipitation, i.e., drizzle and rain drops.

Few global LWP datasets exist and differences in global mean LWP of a factor of two are reported by Lohmann and Neubauer (2018). These findings reflect the different sensing principles, i.e., microwave radiometry and visible/near-infrared techniques. Satellite microwave imagers such as the Special Sensor Microwave Imager (SSM/I) provide LWP estimates for several decades but are limited to the ice free oceans where the background signal is low. LWP is mainly derived from the thermal emission signal in window regions with low water vapor contribution. Microwave receivers also sense rain water within the satellite footprint which can be as large as several tens of kilometer. Recently, the Multisensor Advanced Climatology of Liquid Water Path (MAC-LWP; Elsaesser et al. (2017)) covering the period 1988 to 2016 has been generated. Elsaesser et al. (2017) additionally estimate the contribution of rain water path (RWP) to the total LWP by a simple parametrization and recommend to only use those values with a ratio RWP:LWP of less than 0.2. Greenwald et al. (2018) evaluate MAC-LWP using measurements by the Moderate Imaging Spectroradiometer (MODIS), the CloudSat Profiling Radar (CPR; Stephens et al. (2002)) and the Cloud-Aerosol Lidar with Orthogonal Polarization (CALIOP; Winker et al. (2007)). They found in some cases a net LWP bias of more than 50 percent of the mean CLWP due to the combined effects of the in-cloud and adjacent precipitation biases as well as the cloud-rain partition.

Visible/near infrared techniques as those applied to MODIS exploit the spectral response of reflected sun light to derive LWP from optical depth and effective radius retrievals and are therefore limited to daytime. As the signal mainly relates to the upper part of the cloud, assumptions of the cloud vertical structure introduce uncertainties (Zhou et al., 2016). The horizontal MODIS resolution of about one kilometer is much better than those of microwave satellites. Therefore, MODIS data also have been used to assess the clear sky bias of microwave retrievals (Greenwald et al., 2018), to combine them with microwaves for a better assessment of low clouds (Masunaga et al., 2002), and to detect the ratio of rain and cloud water in low-latitude shallow marine clouds via combination with CPR (Lebsock et al., 2011). In summary, quantifying the accuracy of LWP observations





is a major challenge as no absolute reference exists. While shipborne microwave observations have potential for satellite LWP evaluation (Painemal et al., 2016), they fail during precipitation events, due to a wet radome.

In this study, we use the Next generation Advanced Remote sensing for VALidation studies (NARVAL; Stevens et al. (2019, in press)) expeditions for investigating LWP and its uncertainty in the tropical North Atlantic. The NARVAL missions aim at
improving the understanding of clouds, their role for the distribution of water in the atmosphere, and their interaction with the environment (Bony et al., 2015). Within NARVAL, the German High Altitude and LOng range research aircraft (HALO; Krautstrunk and Giez (2012)) was configured as an airborne cloud observatory combining active and passive microwave instruments with water vapor lidar, solar reflectance measurements and dropsondes. Measurements taken during two campaigns in December 2013 (dry season) and in August 2016 (wet seasons) allow to study clouds with similar, however, more sensitive
and higher spatially resolving instrumentation than available on satellites.

Schnitt et al. (2017) demonstrate the ability of the HALO NARVAL 2013 instrumentation to characterize shallow clouds in the tropical North Atlantic in terms of size, integrated water vapor (IWV), LWP, and surface reaching precipitation using classical regression algorithms. Their study shows sub-footprint variability of spaceborne Special Sensor Microwave Imager/Sounder (SSMIS) and illustrates how MODIS products likely underestimate LWP of thick clouds due to its sensitivity towards the upper
part of the cloud. In this study, we refine the LWP retrieval by making use of high resolution simulations that start to resolve cloud scale circulations and were performed over the full tropical North Atlantic with the ICON (ICOsahedral Non-hydrostatic) weather model to support the analysis (Klocke et al., 2017). We further assess the LWP retrieval accuracy over a wide range of cases, extend the retrieval towards a separation of rain and clouds, and reanalyze the dry season measurements in relation to the wet season campaign.

First, we aim to provide an accurate LWP dataset including uncertainty estimates to support the NARVAL overall goals. For this purpose, we develop retrieval algorithms using multi-channel microwave radiometer measurements as input for LWP and - based on the similar principle - IWV. The novel cloud resolving ICON simulations serve as a training dataset (Sec. 2). In contrast to LWP, IWV can be evaluated using simultaneous measurements by dropsondes and water vapor lidar. The evaluation is presented in Sec. 3. The assessment of LWP (Sec. 4) reveals the importance to use ancillary measurements, i.e., lidar
measurements for low LWP values and cloud radar measurements for lightly precipitating cases. For the latter an RWP retrieval is developed and assessed (Sec. 5). Finally, the campaign data are analyzed to investigate differences between dry and wet season (Sec. 6).

## 2   Material and methods

This section presents the data and methods used in this study. That includes an introduction to the two NARVAL campaigns and
the relevant measurements that were conducted during both campaigns. Furthermore, the generation of the retrieval database and the subsequent retrieval development are presented.



## 2.1 Campaign overview

During the NARVAL expeditions HALO was operated out of Grantley Adams International Airport on Barbados to observe trade wind cumuli and their environment over the tropical North Atlantic (Fig. 1). Different flight patterns were chosen to perform satellite underflights, survey the area, probe the environment of a tropical cyclone, and to determine the large scale

vertical motion by launching several dropsondes within circles of approx. 170 km diameter (Bony and Stevens, 2019, in press). In total eight research flights were performed during NARVAL1-South in December 2013 and ten flights during NARVAL2 in August 2016. NARVAL1 also included research flights in the northern sector of the Atlantic which are not considered here. For simplicity we refer to the southern part as NARVAL1 in the following.

Flight altitudes varied between 6.4 km and 15.0 km with an average speed above ground of $237\,\mathrm{m\,s^{-1}}$ and $207\,\mathrm{m\,s^{-1}}$ during

NARVAL1 and NARVAL2, respectively. All further analyses refer to the area from 37°W to 60°W and 7°N to 20°N. A detailed description of the different research flights can be found in Klepp et al. (2014) for NARVAL1 and Stevens et al. (2019, in press) for NARVAL2.

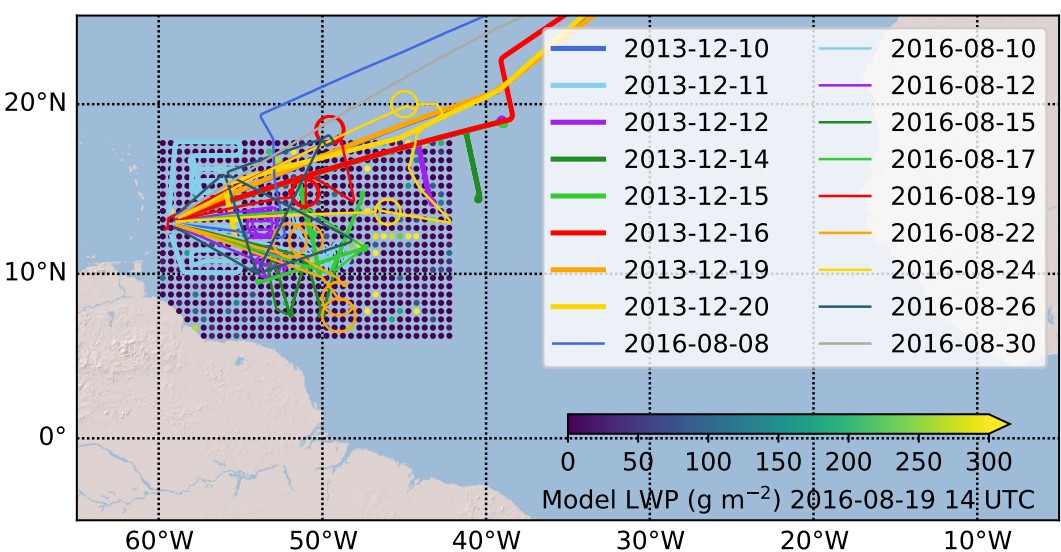

**Figure 1.** NARVAL1 (thin lines) and NARVAL2 (thick lines) flight patterns. The study area of interest is depicted by sub-sampled ICON LWP from August 19, 2016 14:00 UTC. A grid point is shown every 0.5° as present in the retrieval database.

## 2.2 Measurements

The microwave radiometer (MWR) being part of the HALO Microwave Package (HAMP; Mech et al. (2014)) provides the

15 key measurements for this study. HAMP was installed in a belly pod below the HALO fuselage in nadir looking configuration. While the 26-channel MWR includes channels from 22 GHz to 195 GHz, we only use the seven K-band channels (22.24 GHz




to 31.40 GHz) and the 90 GHz channel to retrieve LWP or IWV in the present work. At these frequencies ice particles do not influence the microwave signal substantially with the exception of precipitation sized particles.

As we focus on warm clouds only, cases of ice precipitation are filtered using the differential response of two frequencies along the 60 GHz and 118 GHz oxygen lines. The channels at 53.75 GHz and $118.75 \pm 1.4$ GHz have similar temperature weighting functions but the higher frequency is more affected by ice scattering. The difference between a moving median of differential brightness temperature (BT) to the instantaneous differential BT is used to define the "ice flag". This procedure flags 1.2 % of the measurements of both campaigns.

Both liquid water and water vapor emit microwave radiation across the full microwave spectrum albeit with different spectral sensitivity (Fig. 2). BTs around the 22.235 GHz water vapor rotational line increase with increasing water vapor. The effect is strongest at the line center and decreases along the pressure broadened wing of the absorption line. However, due to water vapor continuum absorption, BTs at window frequencies near 30 GHz and 90 GHz are still affected. In contrast, the influence of liquid water is more dominant in the higher window channels than in absorption channels due to increasing emission with frequency. This can be best seen under low humidity conditions by the increasing BT with increasing frequency.

Figure 2 illustrates the difficulty of retrieving LWP/IWV as in certain channels (e.g., 90 GHz) it is indistinguishable whether BT changes result from changes in IWV or LWP. Therefore, a combination of at least two channels is needed for retrieving IWV or LWP. Note that measurement errors in any of the channels affect both, IWV and LWP retrievals (Crewell and Löhnert, 2003). This means that a good retrieval of either IWV or LWP indicates a good retrieval capability of the other. Thus, an accurate IWV retrieval is a prerequisite of a good LWP retrieval. Note that in most LWP retrievals (e.g., Wentz and Meissner (2000)) the liquid is assumed to consist of cloud droplets only and therefore bulk approaches to calculate the liquid water absorption coefficients are used. However, for rain drops the Rayleigh approximation is not valid anymore and Mie effects need to be considered, though the discrimination of the cloud and rain signal using MWRs is difficult.

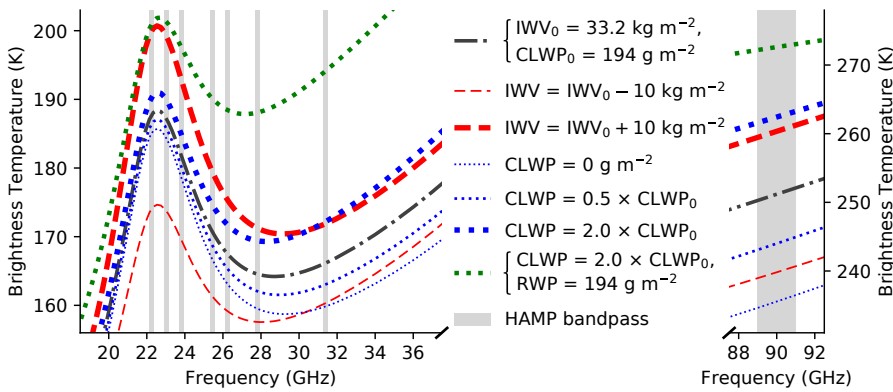

**Figure 2.** Sensitivity of brightness temperatures in the K-band and around 90 GHz to integrated water vapor (IWV), cloud liquid (CLWP), and rain (RWP) water path. Dashed and dotted lines show variations in IVW and LWP, respectively. Bandpasses of the HALO Microwave Package (HAMP) channels are indicated by gray bars. Calculations are based on a thermodynamic dropsonde profile and a synthetic cloud in nadir geometry above the ocean.





The HAMP MWR measures BT with one second integration time and a noise level of less than 0.5 K in the considered channels. Despite ground calibration using hot/cold targets on the air field, BT offsets were identified by comparison with forward simulated dropsondes. Flight dependent corrections were developed (Konow et al., 2018a) and corrected BTs are available in the Climate and Environmental Retrieval and Archive (CERA) (Konow et al., 2018b, c).

HAMP also includes a 35.5 GHz cloud radar with a sensitivity of $-30$ dBZ at 13 km distance in the NARVAL setup. Profiles of the radar reflectivity factor ($Z$) and the linear depolarization ratio are recorded with 30 m vertical and 1 s temporal resolution. To supplement HAMP, Vaisala RD94 dropsondes were launched from HALO to provide the thermodynamic conditions of the environment. In total 76 and 215 sondes were released during NARVAL1 and NARVAL2, respectively.

    To distinguish between clear sky and cloudy conditions as good as possible, Schnitt et al. (2017) derive a cloud mask
for NARVAL1 based on the nadir spectral solar radiance measurements by HALO-SR (HALO Solar Radiation; Fricke et al. (2014)). Unfortunately, sun glint in August deteriorated the cloud mask retrieval during NARVAL2. Therefore, Gödde (2018) developed a cloud mask product using the imaging spectrometer specMACS (spectrometer of the Munich Aerosol and Cloud Scanner; Ewald et al. (2016)) which overcomes the sun glint problem. However, specMACS was not installed during NARVAL1. In order to have similar cloud mask performance during both campaigns the aerosol backscatter profile measured by
the WAter vapor Lidar Experiment in Space (WALES) airborne demonstrator (Wirth et al., 2009) is used instead to provide an along track cloud mask with 1 s resolution.

    WALES also provides profiles of water vapor molecular number density based on the differential absorption lidar (DIAL) principle. These profiles are converted to volume mixing ratio profiles using temperature and pressure data from ECMWF analyses. A resolution of about 200 m vertical and 12 s temporal was chosen as a compromise between accuracy and resolution.
The water vapor data is given on the vertical grid of the raw backscatter data which is 15 m, but smoothed with an averaging kernel of 200 m width (full width at half maximum, FWHM). Water vapor profiles are provided down to about 250 m in cases with no or optically thin clouds, which can be penetrated by the lidar beam. Water vapor information is available below thin clouds, but the cloud itself is masked out in the profile.

    The requirement of simultaneous measurements by all sensors reduces the dataset. While all research flights during NARVAL1 can be used, hardware issues during NARVAL2 prevent having data during some flights as summarized in Tab. 1. The
spatial sampling differs even with the same temporal sampling due to footprints differences. The HAMP MWR has the largest beamwidth in its lowest frequency channel of 5.0° (FWHM). The corresponding surface footprint at 10 km altitude is about 870 m across and 1090 m along track. The HAMP radar beamwidth is 0.6° whereas WALES has a field of view of 1.6 mrad. The respective footprints are about 105 m $\times$ 335 m and 16 m $\times$ 216 m. We reduce the along track sampling differences by averaging
temporally, but the cross track sampling issues remain. This means, a cloud covering a lateral part of the MWR footprint might be missed by the lidar or even the radar. Cross track imagers such as specMACS could be used to assess these issues. However, specMACS was only installed on HALO for NARVAL2 and the detailed analysis of HAMP beam filling is beyond the scope of this study. The problem of different footprints and sensitivities of different NARVAL instruments for cloud masking is illustrated by Stevens et al. (2019, in press).



**Table 1.** Dataset availability. Days of research flights from which the datasets are used for the study of NARVAL1 and NARVAL2, respectively.

| Dataset | NARVAL1 day in December 2013 | NARVAL2 day in August 2016 |
| --- | --- | --- |
| HAMP Radiometer | 10, 11, 12, 14, 15, 16, 19, 20 | 8, 10, 12, 15, 17, 19, 22, 24, 26, 30 |
| HAMP Radar | 10, 11, 12, 14, 15, 16, 19, 20 | 8, 10, 12, 15, 17, 19, 22 |
| Dropsondes | 10, 11, 12, 14, 15, 16, 19, 20 | 8, 10, 12, 15, 17, 19, 22, 24, 26, 30 |
| WALES water vapor | 10, 11, 12, 14, 15, 16, 19, 20 | 10, 12, 15, 17, 19, 22, 24, 26, 30 |
| WALES cloud mask | 10, 11, 12, 14, 15, 16, 19, 20 | 8, 12, 15, 19, 22, 24, 26, 30 |

## 2.3 Retrieval database

Recently, high spatial resolution simulations with the storm resolving ICON model were able to show how resolved convection and its associated circulation interact with and form the larger scale circulation within the Atlantic inter-tropical convergence zone (Klocke et al., 2017). These simulations serve for training and testing retrieval algorithms. The simulations were per-
formed on a triangular grid with a horizontal spacing of about 1.25 km and 75 vertical levels. The simulations cover the area of 4°S to 18°N and 64°W to 42°W. The data was spatially sub-sampled to reduce the computational effort while still covering the variability of atmospheric profiles. To eliminate atmospheric columns with a high degree of correlation, columns are selected on a $0.5° \times 0.5°$ longitude-latitude grid, so that each time step includes 849 cases over the ocean as indicated in Fig. 1. Data from 24 days with hourly outputs each, spanning the period of each campaign are alternately separated into test and training
data. In general, the training and test data excludes cases with ice or LWP greater than $1000\,\mathrm{g\,m^{-2}}$, i.e., 86 % of all profiles over the ocean is used. This limitation is done as our focus is on liquid clouds and their transition to rain. Note that classical satellite algorithms (e.g., Wentz and Meissner (2000)) are trained with an upper LWP limitation of $300\,\mathrm{g\,m^{-2}}$.

Synthetic HAMP measurements, i.e., BTs and radar reflectivity profiles in nadir view, are simulated for each model column based on its thermodynamic profile and hydrometeors (cloud liquid water, rain, cloud ice, snow, and graupel). The Passive and
Active Microwave TRAnsfer code (PAMTRA; Maahn et al. (2015); Cadeddu et al. (2017)) is used. It is configured with 27 output levels to mimic different flight altitudes (6 km to 15 km). The ICON model was set up with a one-moment microphysics scheme (Baldauf et al., 2011). In PAMTRA, cloud and rain particles and their size distributions are described according to the microphysical scheme of ICON and the single scattering properties for each particle are approximated by the Lorentz-Mie theory. Absorption coefficients of atmospheric gases (i.e., oxygen, water vapor, nitrogen) are calculated after Rosenkranz (1998)
with corrections of the water vapor continuum absorption according to Turner et al. (2009) and the line width modification of the 22.235 GHz water vapor line as proposed by Liljegren et al. (2005). The emissivity of the sea ocean surface is calculated by the FAST microwave Emissivity Model version 5 (FASTEM5; Liu et al. (2011)).

To test the realism of the retrieval database, histograms of BTs were compared with their observed counterparts. Joint histograms of an absorption (22 GHz) and a window channel (31 GHZ or 90 GHz) show that the relations between channel pairs





are depicted in the model and observations in the same way (Fig. 3). In clear sky conditions absorption and window channels are highly correlated with both increasing with increasing moisture albeit the increase is less in the window channels. Clear sky cases with low $BT_{31}$ and $BT_{90}$ are visibly as a line of high occurrence and reveal the linear relation between absorption and window channel BTs as a function of IWV. The simulations and measurements show the same relations but differ slightly in

5    terms of the BT combination distribution within this line as the underlying IWV sampling is slightly different. If liquid water clouds occur, the window channel BTs increase compared to clear sky cases (solid lines in Fig. 3). The window channel at 90 GHz has a higher sensitivity towards LWP compared to $BT_{31}$ as it can be seen by the increased LWP line spread. Rainy cases show higher emissions in all channels (dotted lines in Fig. 3). For thick clouds and rain the most liquid sensitive channel (90 GHz) experiences saturation effects with $BT_{90}$ approaching cloud temperatures. The joint histograms reveal the major

10   signals by liquid and water vapor which are exploited within retrieval algorithms. However, multiple influence factors like the exact vertical structure lead to the variability illustrated in Fig. 3. Minor deviations between observations and simulations are visible in the frequency of combinations of $BT_{31}$ and $BT_{90}$ with high $BT_{22}$. Those combinations are associated with heavy precipitation and were observed less frequently than present in the model as flight patterns avoided heaviest precipitation.

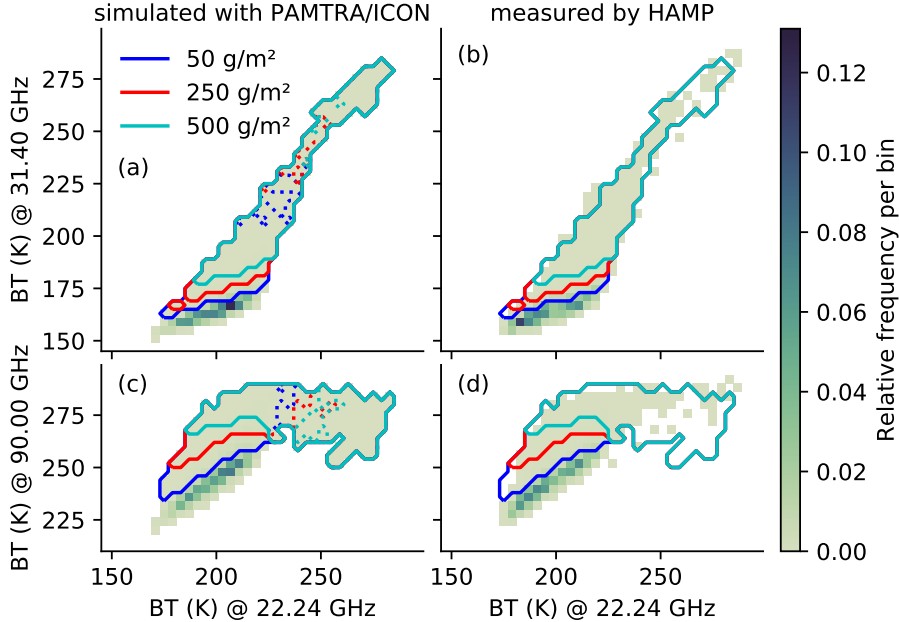

**Figure 3.** Relation between brightness temperatures (BT) in an absorption channel (22.24 GHz) and two window channels, i.e., 31.40 GHz (a, b) and 90.00 GHz (c, d). Two-dimensional histograms of occurrences in simulations (a, c) and HAMP measurements (b, d). Solid contours highlight BT combinations in simulations that mostly occur with LWP higher than 50 g m$^{-2}$, 250 g m$^{-2}$, and 500 g m$^{-2}$. Dotted lines highlight combinations where RWP mostly exceeds the respective threshold. LWP contours in b and d are taken from a and c for guidance. HAMP data from all NARVAL2 flights and ICON/PAMTRA data of the corresponding dates are used. Profiles and measurements with ice are excluded.





## 2.4 Retrieval

The atmosphere emits radiation depending on the atmospheric state as illustrated in Fig. 2. In general, the retrieval of the atmospheric state from MWR measurements is under-determined as multiple atmospheric states can lead to the same set of BTs. Statistical relations have to be established to link the measurement to the most common atmospheric state that can provide

those measurements. To account for non-linearity in this inverse modeling problem, we use an artificial neural network (NN) model similar to Cadeddu et al. (2009) to relate BTs to IWV and LWP. The NN is configured with eight input neurons and 15 hidden neurons in one hidden layer.

For testing and training the retrievals, Gaussian noise of 0.5 K is added to the simulated BTs to account for uncertainties of the HAMP MWR, the radiative transfer, and absorption model. The database is separated by alternating days into test

and training data. Retrieval parameters are derived for each PAMTRA output level to account for the altitude dependence of the microwave signal. The output levels are chosen such, that a HALO flight level never deviates than 90 m from the next output level. The parameters at output levels closest to HALO's altitude are interpolated to HALO's altitude in the retrieval application. Retrieval parameters are derived separately for both campaigns. For testing, each retrieval is applied to the test data of the campaign it is trained for.

In the retrievals, IWV and LWP, and later cloud liquid water path (CLWP) and rain water path (RWP) are the integrals of the water vapor and liquid water over the whole column as seen from space. Despite the fact that HALO flies lower, we chose the total integrals as they prevent artificial flight level depended biases in statistics and allow a comparison with satellite and model data. According to ICON model data, typically less than 0.1 kg m$^{-2}$ water vapor is above a flight altitude of 10 km. About 1 kg m$^{-2}$ of IWV is not seen by the MWR at the lowest NARVAL2 flight altitude of 6.4 km but is included in the retrieval. The

LWP retrieval is trained with the integral of all liquid water, that is given by the model either as cloud water or rain water. The sum of both is used due to the difficulty of MWRs to distinguish clouds and rain (Fig. 2).

Neural network LWP retrievals are compared with linear regression (REG) models as used by Schnitt et al. (2017). The regression relates measured brightness temperatures $BT_i$ to LWP including the quadratic terms of $BT_i$

$$LWP = c + \sum_i \left( b_i\, BT_i + a_i\, BT_i^2 \right),\tag{1}$$

where $a_i$, $b_i$ and $c$ are regression coefficients. Such REGs are less susceptible than NNs to extrapolation towards unforeseen input data, i.e., data values or combinations that are not covered by the training data. However, NNs are better in representing non-linear effects that are apparent in microwave radiative transfer and thus can better adjust to the extremes of the LWP target space. The application of the retrievals to test data reveals overall uncertainties between 0.5 kg m$^{-2}$ and 0.6 kg m$^{-2}$ for IWV for both approaches, i.e. NN and REG, and 22 g m$^{-2}$ and 26 g m$^{-2}$ for LWP using the NN and REG, respectively. For LWP the

uncertainty strongly depends on atmospheric conditions as it will be investigated in Sec. 4.

When retrieval algorithms are applied to HALO measurements, slight biases are observed under clear sky which show slow changes with time. To reduce these biases and to improve the retrieval of low LWP values, we follow the synergistic approach by van Meijgaard and Crewell (2005). Herein, we use the WALES cloud mask for clear sky identification. HAMP measurements





are considered clear sky if no cloud is detected by WALES within $\pm$ 2 seconds flight time. The distance weighted average clear sky LWP within $\pm$ 30 minutes is then subtracted from each a priori retrieved LWP value.

In thick clouds we occasionally observed, that while the REG retrieval gave LWP > $1000\,\mathrm{g\,m^{-2}}$, the NN LWP time series showed a sudden decline. This is likely caused by the clipping of the NN retrieval at $1000\,\mathrm{g\,m^{-2}}$, which is expected as the

retrieval database is limited to LWP < $1000\,\mathrm{g\,m^{-2}}$ and thus BTs associated with higher amounts of liquid are unknown to the retrieval. To avoid this behavior, we use a second NN retrieval trained with an extended database up to $4000\,\mathrm{g\,m^{-2}}$ to flag scenes that are potentially above $1000\,\mathrm{g\,m^{-2}}$. Overall, 0.76 % of the measurements were masked in this way. Note that these measurements often coincide with ice scattering depressions in channels at higher frequencies.

To retrieve the contribution of rain drops (RWP) to the total LWP, the vertically integrated radar reflectivity is used in

addition to the MWR channels in another NN retrieval. The aim is separating the LWP into CLWP and RWP, i.e., splitting the contributions from small cloud droplets and larger rain drops by estimating the fraction

$$f = \frac{RWP}{LWP} = \frac{RWP}{RWP + CLWP}. \tag{2}$$

This retrieval is based on the hydrometeor classes of rain and cloud liquid water in the ICON model. The RWP is calculated by multiplying $f$ and the retrieved total LWP.

## 3   Assessment of integrated water vapor

Three independent methods to derive IWV are available from HALO: the MWR retrieval, vertically integrated humidity from dropsondes, and vertically integrated humidity from WALES. Each of the three methods has its advantages and shortcomings. The microwave radiometry can not provide profile information but gives continuous IWV under nearly all sky conditions. The dropsondes provide in situ measurements, but no valid data up to about the first half kilometer below the aircraft because of the

sensor's adjustment from the aircraft cabin conditions to the outside. Furthermore, wind drifts sondes out of the aircraft nadir with a typical horizontal drift during the decent of 4 km. The dropsonde relative humidity sensor has a repeatability of 2 % according to the manufacturer (Vaisala, 2017). This relates to an IWV accuracy of about $1.4\,\mathrm{kg\,m^{-2}}$. WALES provides water vapor profiles, but they are only available when no cloud extinguishes the laser beam. This limits the application of WALES for the IWV retrieval to clear sky scenes.

To compute the numerical derivative in the DIAL equation, the first data point is at about 250 m above the sea surface and centered at the retrieval interval. Therefore, in the vertical integration, the missing near surface information is filled with the median mixing ratio in the lowest five range bins. The median is chosen to reduce any surface artifacts which can occur, when the first raw data signal point used in the retrieval contains the surface reflex. We estimate that the error of this assumption is about $0.3\,\mathrm{kg\,m^{-2}}$ by analyzing dropsonde humidity profiles. The IWV estimation is discarded if information of more than

400 m above sea-level is missing or there is a gap due to a thin cloud. Also, stability of the estimated WALES IWV is required, which means that the differences to the preceding and succeeding IWV estimations have to be smaller than $2\,\mathrm{kg\,m^{-2}}$.

An example of water vapor retrievals on August 19, 2016 is shown in Fig. 4. An elevated moisture layer between 3 km to 4 km altitude is visible in the first half of the scene. Around 14:53 a plume of moist air reaching up into even higher levels



causes an IWV gradient of nearly $10\,\mathrm{kg\,m^{-2}}$ ($26\,\mathrm{kg\,m^{-2}}$ to $35\,\mathrm{kg\,m^{-2}}$) over a distance of about $110\,\mathrm{km}$. This gradient is captured well by WALES and HAMP. The two dropsondes that were released between 14:45 and 14:55 reconstruct this gradient, but both have a dry offset. This offset might be due to drifting of the sonde towards the drier air mass. After a short outage of WALES at around 15:00, shallow clouds below $2\,\mathrm{km}$ prevent the determination of lidar IWV frequently. Most of the IWV

measurements from dropsondes agree with the coincident remote sensing estimates within the sondes' uncertainty.

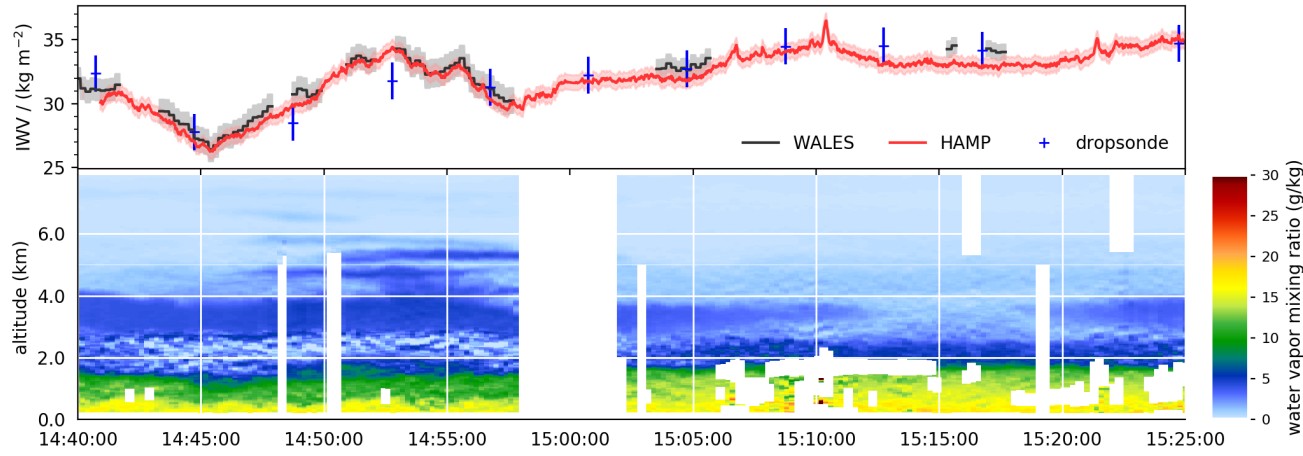

**Figure 4.** Water vapor time series from NARVAL2 research flight 6 on August 19, 2016. Top: IWV time series by HAMP, WALES and dropsondes with their uncertainties. Bottom: WALES water vapor profiles. White areas denotes masked lidar data. The scene represents a circle around 14.8°N and 51.0°W over a distance of 520 km.

A more quantitative comparison is achieved by considering all measurements from both campaigns which cover a wide variety of water vapor conditions (Fig. 5). Overall, the sondes agree well with HAMP over the whole observed range from very low ($20\,\mathrm{kg\,m^{-2}}$) to very high ($60\,\mathrm{kg\,m^{-2}}$) values of IWV (Fig. 5a). The root-mean-square deviation (RMSD) is $1.39\,\mathrm{kg\,m^{-2}}$ ($1.28\,\mathrm{kg\,m^{-2}}$) with a mean bias of $0.28\,\mathrm{kg\,m^{-2}}$ ($0.47\,\mathrm{kg\,m^{-2}}$) during NARVAL1 (NARVAL2). The positive biases of HAMP is

most likely caused by the retrieval, which is trained with the whole column IWV, whereas the sonde IWV is only integrated along its measurement path. Most sondes were released above $9\,\mathrm{km}$ which would miss an IWV of about $0.2\,\mathrm{kg\,m^{-2}}$ according to ICON data. Note that dropsondes released from below $6.5\,\mathrm{km}$ are discarded in the comparison to avoid an artificial bias. The random error between HAMP and sondes ($1.2\,\mathrm{kg\,m^{-2}}$) is smaller than the estimated uncertainties of the dropsonde ($1.4\,\mathrm{kg\,m^{-2}}$) and the MWR retrieval ($0.6\,\mathrm{kg\,m^{-2}}$) which indicates the high quality of the measurements as uncertainties due to spatio-

temporal mismatch are included in the RMSD as well. Note that uncertainties due to MWR calibration are largely compensated as offsets between measured BT from those derived by radiative transfer calculations based on dropsondes have been corrected by Konow et al. (2018a).

WALES IWV can be used for continuous comparison to HAMP IWV along the flight track in clear sky scenes. A comparison of all coincident measurements during NARVAL2 is depicted in Fig. 5b. The average bias between HAMP and WALES IWV





is $-0.59\,\mathrm{kg\,m^{-2}}$. The bias is cut in half when considering only the 40 simultaneous measurements during which a dropsonde was launched (Tab. 2). The random error is smaller in contrast to the HAMP – dropsonde comparison. This is likely due to the better spatial match between the two nadir measurements compared to a drifting sonde. However, higher RMSD between HAMP and WALES IWV can be found during NARVAL1, which is mostly related to a higher bias. The bias increases to

$1.70\,\mathrm{kg\,m^{-2}}$ in the HAMP – WALES comparison when only considering measurements during which a sonde was released. A bias of similar magnitude is apparent between WALES and the dropsondes. Most likely the dry bias of WALES is due to the method of how the 12 s water vapor profiles are derived. The profiles only contain raw profiles (within the 12 s), that are not blocked by a cloud. For small scale boundary layer convection, this means preferred sampling of downdraft regions. In these downdraft regions dry air is entrained from the rather dry free troposphere into the convection layer during NARVAL1 (Stevens

et al., 2017). This results in biased sampling of rather dry profiles. During NARVAL2 humidity was reaching higher altitudes, which resulted in less entrainment of dry air in cloud gaps.

    With the exception of the HAMP – WALES comparison during NARVAL1, the RMSD between the different instrument pairs is found between $0.8\,\mathrm{kg\,m^{-2}}$ and $1.4\,\mathrm{kg\,m^{-2}}$ (Tab. 2). This corresponds to an error of 2 % to 7 % over the observed range of $20\,\mathrm{kg\,m^{-2}}$ to $60\,\mathrm{kg\,m^{-2}}$. For comparison, Mears et al. (2015) found random IWV deviations between different spaceborne

MWR and ground-based GPS (Global Positioning System) instruments of $1.7\,\mathrm{kg\,m^{-2}}$ to $2.0\,\mathrm{kg\,m^{-2}}$ over a similar IWV range using 26 small island stations located mainly in the tropics.

    The HAMP IWV retrieval has a theoretical uncertainty of about $0.6\,\mathrm{kg\,m^{-2}}$ which is derived by applying the IWV retrieval to simulated measurements from the test database (Sec. 2.4) and is constant over a wide IWV range (not shown). This is well in line with the RMSD derived in the pairwise comparisons taking into account the estimated uncertainties of WALES and

dropsondes as well as uncertainties due spatio-temporal mismatch. In summary, the pairwise comparisons in relation to the individual uncertainties indicate high HAMP IWV performance and the suitability of our retrieval approach.

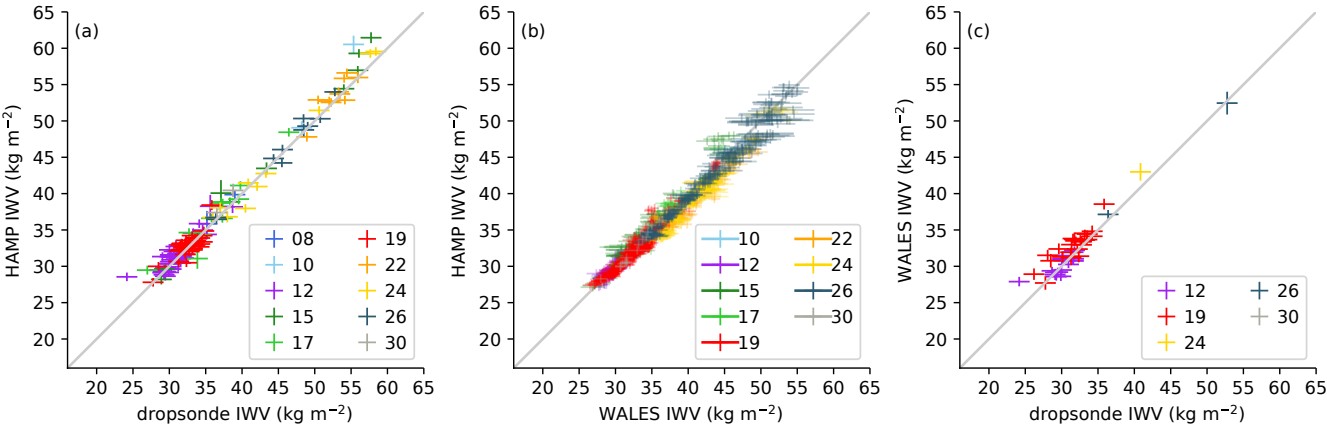

**Figure 5.** IWV comparison of HAMP MWR and dropsondes (a), HAMP MWR and WALES (b) and WALES and dropsondes (c) during NARVAL2. The colors indicate the flight days in August 2016. Scores are given in Tab. 2.



**Table 2.** Comparison of IWV retrieved from HAMP, WALES and dropsondes. Pairwise observations of two instruments and the subsets of the observations for that all instruments were available. Bias, root-mean-square deviation (RMSD), and bias corrected RMSD (STD) in kg m$^{-2}$.

|  |  | HAMP - sondes | HAMP - WALES | WALES - sondes |
|---|---|---|---|---|
| NARVAL1 observed pairwise | bias | 0.28 | 0.92 | −1.21 |
|  | RMSD | 1.39 | 1.36 | 1.60 |
|  | STD | 1.38 | 1.01 | 1.07 |
|  | count | 43 | 2482 | 24 |
| NARVAL1 observed by all | bias | 0.32 | 1.70 | −1.37 |
|  | RMSD | 1.21 | 2.20 | 1.70 |
|  | STD | 1.20 | 1.41 | 1.03 |
|  | count | 21 | 21 | 21 |
| NARVAL2 observed pairwise (Fig. 5) | bias | 0.47 | −0.59 | 0.73 |
|  | RMSD | 1.28 | 1.21 | 1.38 |
|  | STD | 1.19 | 1.06 | 1.19 |
|  | count | 146 | 1632 | 47 |
| NARVAL2 observed by all | bias | 0.32 | −0.25 | 0.57 |
|  | RMSD | 1.16 | 0.82 | 1.23 |
|  | STD | 1.12 | 0.79 | 1.11 |
|  | count | 40 | 40 | 40 |

## 4 Assessment of liquid water path

There are no independent measurements of sufficient quality to assess the quality of the LWP product. However, the large retrieval database (173 339 ice free cases in the test dataset) allows a theoretical in-depth analysis of the retrieval performance. This approach is supported by the good consistency between the BTs in the database and the HAMP measurements in terms of relation resemblance (Fig. 3) and performance of IWV retrieval (Sec. 3). We analyze the retrieval error as a function of the true LWP as well as of the retrieved LWP using the database.

First, we analyze the difference of retrievals developed with all ice free cases of the training database (all sky) and with cloudy cases only, which reduces the dataset size to about one quarter. A model profile is regarded as cloudy if LWP > 1 g m$^{-2}$. REG and NN retrievals are trained with the all sky and the cloudy sky datasets separately. The errors of retrieved LWP from the test database are calculated for bins of the true LWP. Both REG and NN show similar behavior of the RMSD between the retrieved LWP and the model truth with increasing LWP. The RMSD is constant for LWP below about 30 g m$^{-2}$ and increases



with LWP, e.g. $50 \, \mathrm{g \, m^{-2}}$ at $500 \, \mathrm{g \, m^{-2}}$. For LWP values $> \sim 800 \, \mathrm{g \, m^{-2}}$ the number of test cases reduces strongly leading to less robust results. For LWP $< 30 \, \mathrm{g \, m^{-2}}$, the errors are smaller for REG and NN retrieval types if the clear sky cases are included in the training (compare Fig. 6a and 6b). Including clear sky in the training, the retrieval errors decrease slightly for a REG model and are almost cut to half for an NN. This shows the ability of an NN to non-linearly relate a variety of BT combinations

to zero LWP. However, retrievals that are especially trained for all sky scenes have a considerably larger RMSD for LWP $> 20 \, \mathrm{g \, m^{-2}}$ than those trained with cloudy cases only as clear sky cases make up $77 \, \%$ of the data. Since we are targeting clouds and not clear sky, we decided for a retrieval trained with data excluding the clear sky model profiles. Instead, we make use of lidar measurements which are better suited than MWR for cloud masking.

Regarding cloudy sky retrievals, the RMSD for a given (true) LWP less than $40 \, \mathrm{g \, m^{-2}}$ is smaller when using the NN retrieval
instead of a REG model (Fig. 6a). This is related to a suppression of unphysical negative LWP values by the NN. Thus, in contrast to a REG which has a nearly Gaussian error characteristic, the NN tends to overestimate LWP. This results in a more negative mean LWP error (true - retrieved) of clouds with less than $10 \, \mathrm{g \, m^{-2}}$ but also in a smaller interquartile range of errors when using the NN instead of the REG. However, the retrieval error for true LWP $< 10 \, \mathrm{g \, m^{-2}}$ remains in the order of $10 \, \mathrm{g \, m^{-2}}$ to $18 \, \mathrm{g \, m^{-2}}$ even when using the NN.

The bias errors visible in Fig. 6a can not be used to adjust the retrieved LWP as the true LWP value is not known in practice. For the application of the error analysis on measurements, it is important to analyze the LWP error as a function of the retrieved LWP. The RMSDs of the NN and REG retrievals are larger than $100 \, \%$ for a retrieved LWP below $12 \, \mathrm{g \, m^{-2}}$ which can be regarded as a detection limit (Fig. 7). Therefore, ancillary measurements with higher sensitivity are needed to detect these thin liquid water clouds. The RMSD is below $20 \, \mathrm{g \, m^{-2}}$ for REG LWP $< 50 \, \mathrm{g \, m^{-2}}$ and NN LWP $< 100 \, \mathrm{g \, m^{-2}}$, and moderately
increases with increasing LWP. Therefore, the relative RMSD decreases from $50 \, \%$ for a retrieved LWP of about $40 \, \mathrm{g \, m^{-2}}$ to $20 \, \%$ for LWP $> 100 \, \mathrm{g \, m^{-2}}$ for both retrieval types. While the RMSD is rather similar for REG and NN, the NN succeeds in capturing the nonlinear retrieval providing a nearly zero bias across the full LWP range and is therefore preferred over REG.

Analyzing the retrieved LWP distribution for clear sky scenes is a widely used method to assess an LWP retrieval (e.g., Liu et al. (2001), Greenwald et al. (2018)) because this characterization can be made from measurements using ancillary
observations that define clear sky scenes. We use WALES measurements for cloud and clear sky indication. The distributions of LWP from HAMP MWR are depicted in Fig. 8 for observed clear sky scenes (blue lines) along the track for both campaigns. The distributions are compared to the theoretical ones of retrieved LWP from all clear sky (true LWP $< 1 \, \mathrm{g \, m^{-2}}$) cases of the respective campaign in the ICON/PAMTRA database (orange lines in Fig. 8). The latter distributions are closely related to the retrieval uncertainty of the lowest LWP bin in Fig. 6a as this represents the retrieval uncertainty for true LWP $< 2.5 \, \mathrm{g \, m^{-2}}$.
The distributions roughly resemble Gaussian behavior with mean values of about $10 \, \mathrm{g \, m^{-2}}$ and widths of about $9 \, \mathrm{g \, m^{-2}}$. Some differences between NARVAL1 and -2 exist which are even stronger for the measured distributions. During NARVAL1, the measured distribution is skewed towards higher values. This might be caused by cloud patches that were only present in a lateral part of the MWR footprint such that the scene was falsely identified as clear sky by the lidar, which only slices though the center of the MWR footprint. As this effect is not visible for NARVAL2 measurements, it might be that clouds were generally
smaller and more frequent during NARVAL1 (see Sec 6).



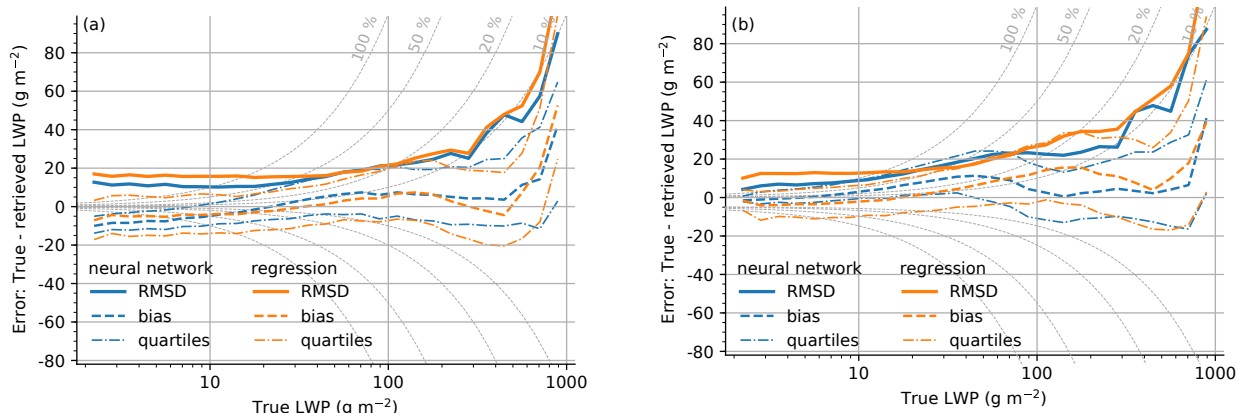

**Figure 6.** Expected retrieval error as function of true LWP for neural network and linear regression LWP retrievals. Retrievals (a) trained for $1\,\mathrm{g\,m^{-2}} < \mathrm{LWP} < 1000\,\mathrm{g\,m^{-2}}$. Retrievals (b) trained including clear sky cases ($\mathrm{LWP} < 1000\,\mathrm{g\,m^{-2}}$). Error measures (colored lines) for logarithmically distributed bins with ten bins per LWP power of ten. Gray dashed lines denote the corresponding relative LWP error.

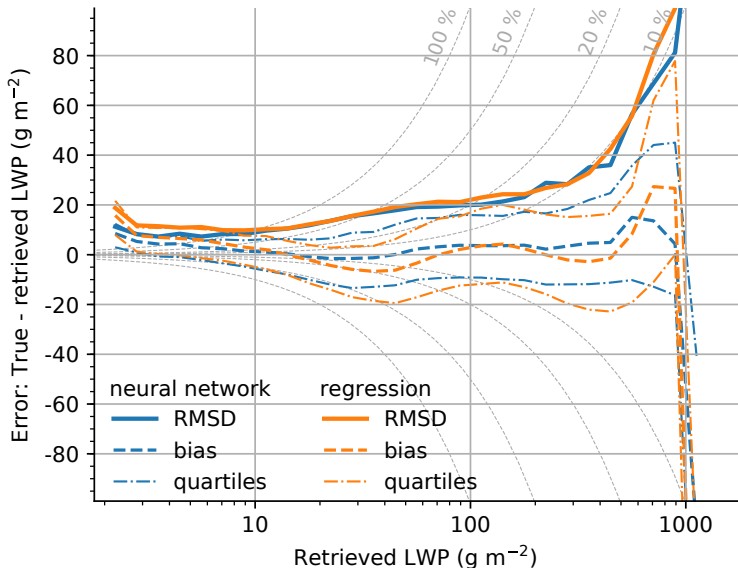

**Figure 7.** As Fig. 6, but with errors shown as function of retrieved LWP. Retrievals are trained and tested with $1\,\mathrm{g\,m^{-2}} < \mathrm{LWP} < 1000\,\mathrm{g\,m^{-2}}$. First bin contains all data with retrieved $\mathrm{LWP} < 2.5\,\mathrm{g\,m^{-2}}$ (including negative).

For both campaigns the similar widths and standard deviations of the retrieved LWP indicate a good agreement between simulations and measurements for clear sky (Fig.8). The apparent second mode at $20\,\mathrm{g\,m^{-2}}$ in the observed clear sky LWP distribution during NARVAL2 is caused by different mean deviations during different flights, probably influenced by the cali-





bration. Overall, the narrow Gaussian widths (11.4 g m$^{-2}$ and 8.3 g m$^{-2}$ for NARVAL1 and -2) of the retrieved clear sky LWP distributions demonstrate the good performance of HAMP compared to evaluation studies by Liu et al. (2001) (28 g m$^{-2}$, airborne) and Greenwald et al. (2018) ($\sim$ 30 g m$^{-2}$, satellite). The better HAMP performance is likely due to its smaller footprint, additional frequency channels, and more recent technology. The sensor synergy of using the lidar cloud mask for clear sky bias

correction (Sec. 2.4) reduces the bias in clear sky conditions to values barely above zero as small cloud patches can still be in the outer area of the MWR footprint which is not transected by the lidar beam. The bias correction further narrows the clear sky LWP distributions. Note that a good agreement (small bias) is expected as the lidar cloud mask is also used to define clear sky. The deviations of the observed clear sky LWP distributions from delta distributions are due to the moving window in the bias correction.

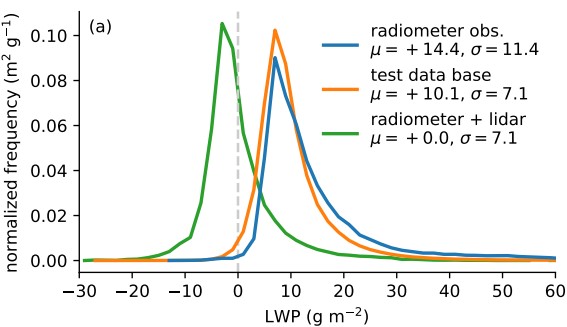
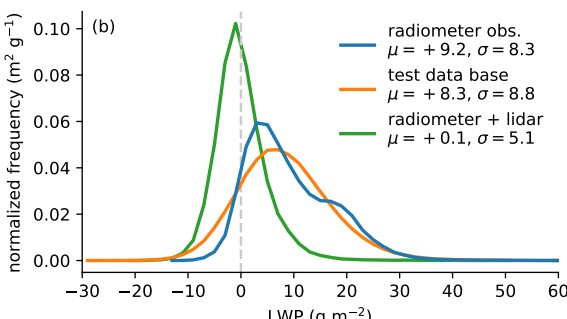

**Figure 8.** LWP distributions retrieved during clear sky scenes only. Shown are the LWP retrieved with the neural network based on radiometer observations (blue lines) during (a) NARVAL1 and (b) NARVAL2, the retrieved LWP from the test database (orange lines) for profiles with LWP < 1 g m$^{-2}$, and the observed LWP after applying the lidar clear sky correction (green lines). Mean ($\mu$) and standard deviation ($\sigma$) are given for each distribution in g m$^{-2}$.

In summary, the ICON/PAMTRA database allows estimating the expected uncertainty of the LWP retrieval. This reveals a lower retrieval limit of about 12 g m$^{-2}$ and an RMSD below 20 g m$^{-2}$ for LWP below 100 g m$^{-2}$ and below 20 % above for the NN retrieval. A narrow clear sky distribution of HAMP measurements (STD $\sim$ 10 g m$^{-2}$) is found that is in good agreement with the theoretical assessment, but a small bias in the order of 12 g m$^{-2}$ remains which is eliminated by the clear sky correction. The synergy of MWR and lidar removes the bias and reduces the clear sky LWP noise to 5 g m$^{-2}$ to 7 g m$^{-2}$.

# 5   Assessment of rain

To investigate the formation of rain with HAMP measurements, this section extends the applicability of the LWP retrieval to drizzle and light precipitation by combining cloud radar with MWR. As described in Sec. 2.4, RWP is retrieved as the fraction $f = \frac{RWP}{LWP}$ by a NN using eight BTs and integrated radar reflectivity as input. Two physical effects are considered in the retrieval: hydrometeor scattering, which becomes more important with increasing droplet size and microwave frequency,





and radar backscatter being sensitive to $D^6$, where $D$ is the droplet diameter. The first effect is considered by including the 90 GHz channel as proposed by Cadeddu et al. (2017). For the latter effect, the vertically integrated (linear) radar reflectivity is used as retrieval input in addition to the MWR channels also used in the LWP retrieval. This integrated reflectivity as a columnar quantity is more comparable to a BT and less noisy than the reflectivity of a single range gate and is thus used as

retrieval input.

The Gilbert skill score (GSS) (Hogan et al., 2010), also known as equitable threat score (ETS), is used to rate how well retrieval "yes" events correspond to true "yes" events while accounting for hits due to chance. "Yes" events mark RWP above a given threshold. The GSS is defined as

$$GSS = \frac{hits - hits\_by\_chance}{hits + misses + false\_alarms - hits\_by\_chance} \tag{3}$$

using the common entries of the contingency table and the hits due to chance

$$hits\_by\_chance = \frac{(hits + misses) \times (hits + false\_alarms)}{hits + misses + false\_alarms + correct\_negatives}. \tag{4}$$

$GSS$ ranges from $-\frac{1}{3}$ to 1 with 1 being the perfect score.

The retrieval of RWP is evaluated for different RWP thresholds (Fig. 9). GSS shows good performance being higher than 0.75 for RWP thresholds from $10\,\mathrm{g\,m^{-2}}$ to about $50\,\mathrm{g\,m^{-2}}$ and higher than 0.5 for RWP up to $250\,\mathrm{g\,m^{-2}}$. Note that 762, 295,

and 62 of the test cases have RWP greater than $10\,\mathrm{g\,m^{-2}}$, $50\,\mathrm{g\,m^{-2}}$, and $250\,\mathrm{g\,m^{-2}}$, respectively and only few samples with higher RWP are available. The hit rate is higher than $80\,\%$ for thresholds between $10\,\mathrm{g\,m^{-2}}$ and $250\,\mathrm{g\,m^{-2}}$, but the $250\,\mathrm{g\,m^{-2}}$ threshold also generates $37\,\%$ false alarms. Especially, the high GSS for low RWP thresholds makes the $f$ retrieval a useful tool combining cloud radar and MWR for detecting measurements, that contain warm precipitation.

A case study of two showering shallow cumuli is shown in Fig. 10 to illustrate the capabilities of retrieving LWP and RWP.

Both clouds contain a precipitating core with maximum RWP of probably more than $200\,\mathrm{g\,m^{-2}}$. The RWP shows stronger relative gradients than LWP (=CLWP + RWP), which indicates a narrow precipitating core. Note that higher horizontally resolved information by radar (MWR footprints 3.3° to 5.0° vs. radar footprint 0.6°) is integrated in the RWP retrieval compared to the LWP retrieval. The RWP retrieval consistently indicates no rain except for the time when the radar signal touches the surface or when there is a clearly visible fallstreak (17:42:30). The two showering clouds reveal maximum LWP of $716\,\mathrm{g\,m^{-2}}$

and more than $1000\,\mathrm{g\,m^{-2}}$, respectively. The latter is likely a lower limit as the LWP retrieval sets the clipping flag. This case study also demonstrates the higher sensitivity of the lidar and the LWP retrieval which detect clouds between 17:38:30 and 17:39:10 which are to thin to be visible in the radar data.




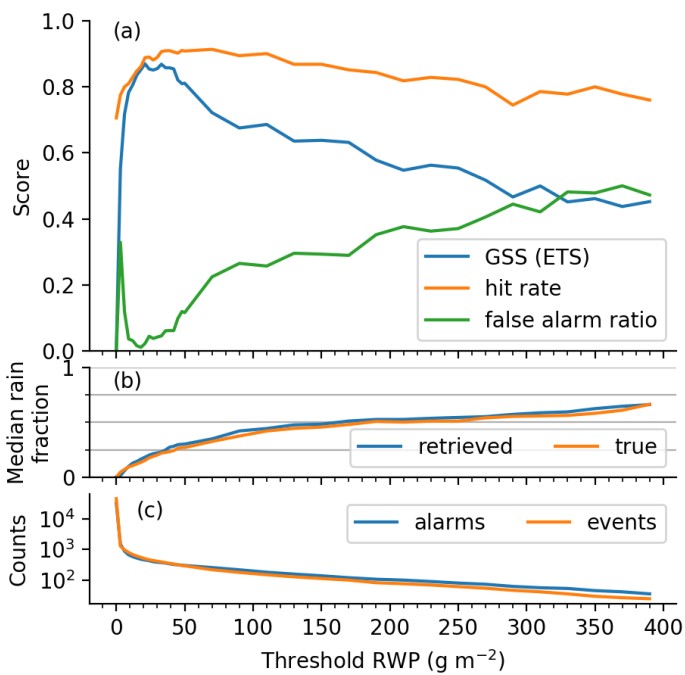

**Figure 9.** Scores for detecting an atmospheric profile with RWP higher than the respective threshold. (a) Gilbert skill score (GSS), hit rate and false alarm ratio. (b) Median fraction of rainwater as a function of RWP threshold. (c) Number of alarms and events for retrieved and true RWP above the threshold, respectively.

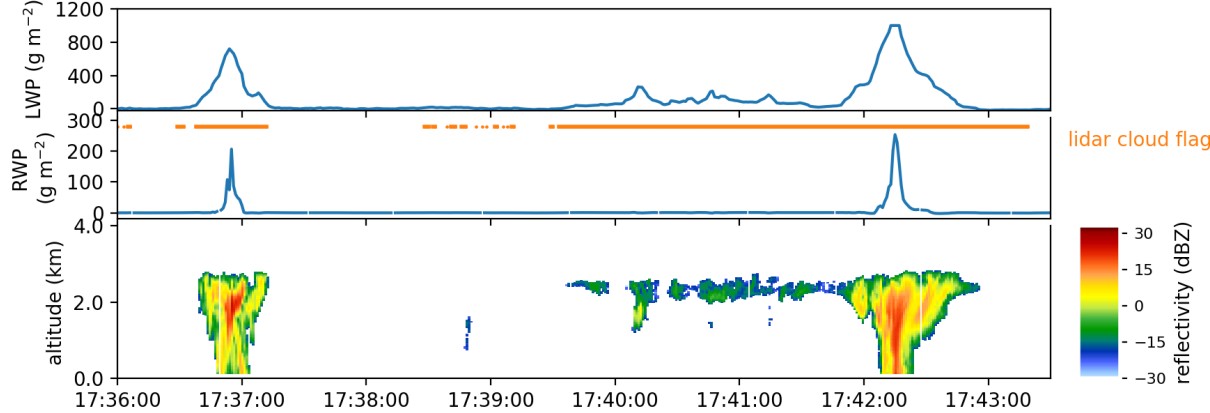

**Figure 10.** Example scene of rain retrieval from NARVAL1 research flight 8 on December 20, 2013. Retrieved LWP (top), retrieved RWP (middle) and radar reflectivity profile (bottom). The scene has a length of 117 km.





## 6 Comparison of dry and wet season

The synergy of lidar, radar and MWR is necessary to understand the difference of clouds in the dry and wet season as all instruments have their specific limitations. The lidar cloud mask indicates the more frequent occurrence of clear sky during the wet season (70.0 %) compared to the dry season (53.3 %, Tab. 3) even though the environment is characterized by less humid air in the dry season (Fig. 11a, b). The IWV distribution is clearly confined to moderate values with a mean of $28\,\text{kg}\,\text{m}^{-2}$ in the dry season which is mainly due to a rather dry middle troposphere seen in the lidar water vapor profiles. During the wet season, IWV values up to $60\,\text{kg}\,\text{m}^{-2}$ were observed distributed into two modes around $35\,\text{kg}\,\text{m}^{-2}$ and $52\,\text{kg}\,\text{m}^{-2}$. These two modes are clearly distinct from the single mode observed in the dry season and reveal the expected humid character of the wet season. The most humid air during NARVAL1 was sampled in a deep convective system on the southernmost leg of research flight 2 on December 11, 2013. This was the NARVAL1 flight during which HALO was closest to the inter-tropical convergence zone (ITCZ). The NARVAL2 IWV distribution seems to be driven by the vicinity of the flight track to deeper convective systems and the ITCZ as it can be analyzed from satellite images and thus also by the selection of flight patterns.

Interestingly, clouds show a higher mean LWP of about $63\,\text{g}\,\text{m}^{-2}$ in the dry season compared to a mean LWP of $45\,\text{g}\,\text{m}^{-2}$ in the wet season. Likewise, thicker clouds (LWP $> 50\,\text{g}\,\text{m}^{-2}$) were more frequent in the dry season (Fig. 11c, d), i.e., 27.1 % of the time when a cloud was seen in the dry season it contained LWP $> 50\,\text{g}\,\text{m}^{-2}$, whereas only 18.6 % of the time in the wet season clouds exceeded this value. However, the variability between flights (Fig. 11) is high and illustrates sampling issues due to flight planing considerations. An analysis of ground-based LWP measurements at the Barbados Cloud Observatory (Stevens et al., 2016) over the years 2013-2018 confirms the generally higher LWP values during December than August (not shown).

The dry season clouds tend to produce light precipitation more frequently than the wet season clouds as indicated by the more frequent exceedance of RWP thresholds (Tab. 3). The cumulative distributions of RWP occurrences of all cloudy measurements with LWP $> 50\,\text{g}\,\text{m}^{-2}$ are depicted in Fig. 11e and f for each flight in the two seasons, when radar measurements are available. The vast majority (NARVAL1: 91 %; NARVAL2: 96 %) of all these measurements show RWP $< 10\,\text{g}\,\text{m}^{-2}$. Higher amounts of light rain seem to be more frequent in the dry season dataset, although the small number of heavy RWP observations inhibits a statistical sound statement as RWP $> 100\,\text{g}\,\text{m}^{-2}$ was only observed for 162 s and 49 s in the radar-radiometer datasets of dry and wet season, respectively. These time spans exclude measurements flagged as clipping (LWP $> 1000\,\text{g}\,\text{m}^{-2}$) or frozen precipitation (ice scattering). While warm precipitation seems to occur less often, clouds associated with frozen precipitation were more often observed in the wet season (1.6 % of the time) than in the dry season (0.5 %). Therefore, the lower LWP of the wet season clouds might be due to a higher precipitation efficiency compared to the dry season.





**Figure 11.** Frequency distribution functions of IWV during the dry season (NARVAL1, a) and wet season (NARVAL2, b), LWP during NARVAL1 (c) and NARVAL2 (d), and cumulative distribution functions of RWP during NARVAL1 (e) and NARVAL2 (f). Colors denote the day in the month of the respective study. Colors in (a) and (b) denote the contribution of each flight to the total distribution. The bin edges are represented as minor ticks in (c) to (f). LWP distributions only include measurements where the lidar cloud flag reports a cloud within $\pm$ 2 seconds. RWP distribution is based on non-clear-sky-corrected LWP dataset (see note [a] Tab. 3), where LWP > 50 g m$^{-2}$.





**Table 3.** Comparing NARVAL1 and NARVAL2 cloud properties observe south of 20°N and with HALO altitude above 6 km. Percentages of flight time with available corresponding datasets during each study.

| Dataset | NARVAL1 December 2013 radiometer, lidar and radar | NARVAL2 August 2016 radiometer and lidar | radiometer and radar |
|---|---|---|---|
| Clear sky | 53.31 % | 69.95 % | – |
| LWP > 20 g m$^{-2}$ | 21.62 % | 10.60 % | – |
| LWP > 50 g m$^{-2}$ | 12.63 % | 5.26 % | – |
| LWP > 500 g m$^{-2}$ | 1.18 % | 0.33 % | – |
| $\overline{\text{LWP}}$ of clouds | 63 g m$^{-2}$ | 40 g m$^{-2}$ | – |
| RWP [a] > 10 g m$^{-2}$ | 1.85 % | – | 0.30 % |
| RWP [a] > 50 g m$^{-2}$ | 0.43 % | – | 0.78 % |
| Ice-flag | 0.51 % | 0.94 % | 1.76 % |
| LWP clipping | 0.97 % | 0.45 % | 0.53 % |
| Total hours | 25:26:18 | 39:43:28 | 41:22:48 |

[a] Based on non-clear-sky-corrected LWP as radar and lidar cloud mask were only available during 5 of 10 flights during NARVAL2.

# 7 Summary and conclusions

Clouds play a critical role in the development of the future climate and especially marine low level clouds have been identified as source of uncertainty. An important cloud macrophysical quantity is LWP. Global observations are limited by satellite resolution or accuracy and ground-based observations over the oceans are few. To fill this observational gap, the NARVAL

studies were initiated to assess North Atlantic trade wind clouds using the HALO research aircraft. We use a multichannel microwave radiometer, a cloud radar, a lidar and a dropsonde system deployed to HALO to provide insights into clouds on the kilometer scale. For NARVAL1 (December 2013) and NARVAL2 (August 2016) a unique retrieval training and test database was developed based on ICON simulations with 1.25 km grid spacing. The database contains more than 350 000 physically consistent profiles that characterize the thermodynamic state of the atmosphere and the hydrometeor distributions during each

of the two campaigns. Synthetic HAMP measurements in terms of BTs and radar reflectivity profiles in nadir view were simulated for each profile using PAMTRA. The synthetic BT measurements show bivariate relations that are consistent to those observed and therefore give trust that the database can be used to develop retrievals and assess LWP quality.

To estimate IWV, LWP, and RWP from HAMP measurements, artificial neural networks are trained with the retrieval database. BTs of seven K-band and the 90 GHz channel are used for IWV and LWP; vertically integrated radar reflectivity

is used in addition for RWP.



Similar to LWP, a IWV retrieval is based on the spectral of BT characteristics between the same water vapor absorption and window channels. A good retrieval of either IWV or LWP is a prerequisite for the other. The IWV comparison to dropsonde measurements and the continuous along-track comparison to the water vapor lidar WALES shows good agreement with an RMSD smaller than $1.4\,\mathrm{kg\,m^{-2}}$ and no distinct error dependence of IWV itself. Overall, the IWV assessment shows the good

practical performance of HAMP and the suitability of the ICON/PAMTRA database for developing microwave retrievals for NARVAL1 and NARVAL2.

LWP retrievals are theoretically assessed as functions of retrieved LWP and true LWP. A slight advantage of the neural network compared to a linear regression retrieval is evident, especially at the limits of the LWP range ($1\,\mathrm{g\,m^{-2}}$ to $1000\,\mathrm{g\,m^{-2}}$). Both approaches show relative errors greater than $100\,\%$ for a retrieved LWP $< 12\,\mathrm{g\,m^{-2}}$, which can be regarded as detection

limits. If more liquid water is contained in the column, the random error decreases to $20\,\%$ at LWP $\approx 100\,\mathrm{g\,m^{-2}}$ and $10\,\%$ at LWP $\approx 800\,\mathrm{g\,m^{-2}}$. Both retrievals show an offset error smaller than the random component for LWP $< 10\,\mathrm{g\,m^{-2}}$ with different signs depending on whether it is analyzed as function of true or retrieved LWP. Because of the ambiguity of the error sign, we conclude that this bias can not be accounted for with the MWR retrieval alone and we developed a synergistic clear sky offset correction using the WALES lidar cloud mask. The cloud mask reduces the noise of clear sky LWP to $7.1\,\mathrm{g\,m^{-2}}$ and $5.0\,\mathrm{g\,m^{-2}}$

for NARVAL1 and NARVAL2, respectively.

To allow investigating the onset of precipitation, a neural network retrieval is trained to estimate the fraction between RWP and LWP from a combination of integrated radar reflectivity factors and BTs. Using the test database, a Gilbert skill score above 0.75 is found for RWP thresholds between $10\,\mathrm{g\,m^{-2}}$ and $50\,\mathrm{g\,m^{-2}}$ which shows good applicability for detection of rain or drizzle onset.

We used data from 36 flight hours in December 2013 (dry season, NARVAL1) and 64 flight hours in August 2016 (wet season, NARVAL2) to investigate differences between the seasons. The analysis shows that although clouds were more frequent and their LWP and RWP were higher during the flights in the dry season, more microwave scattering of ice was observed in the wet season indicating strong precipitation events. The difference between $\overline{\mathrm{LWP}}_{\mathrm{dry\ season}} \approx 63\,\mathrm{g\,m^{-2}}$ and $\overline{\mathrm{LWP}}_{\mathrm{wet\ season}} \approx 40\,\mathrm{g\,m^{-2}}$ is clearly larger than the LWP retrieval uncertainty. As expected, the IWV histograms reveal the dry season as dryer and more

uniform, and the wet season as more humid. However, the IWV distributions also reveal sampling biases due to flight track choices especially for the wet season. Therefore, the airborne measurements need to be combined with long-term ground-based and spaceborne measurements to draw statistically sound conclusions. The fine scale airborne microwave observations such as the measurements obtained with HAMP can be used to investigate the sub-satellite-footprint inhomogeneity of LWP and rain for a better error characterization of satellite measurements.

The synergy of active and passive microwave observations could further benefit from using an optimal estimation approach including the full radar profile and all MWR channels to improve the partition of rain and cloud droplets and frozen particles (e.g., Battaglia et al. (2016)). An extension of the NARVAL observations is planned by the EUREC$^4$A field study in early 2020 ("elucidate the couplings between clouds, convection and circulation"; Bony et al. (2017)) and the algorithms presented here will be applied. This campaign will provide additional observations of large scale dynamics and horizontal remote sens-





ing observations by a second aircraft in the cumulus layer. Together with ship measurements a unique dataset for a better understanding of precipitation onset will be generated.

*Data availability.* The time series of IWV, LWP, and RWP are available in the CERA (Jacob et al., 2019a, in prep., b). The HAMP MWR, HAMP radar and dropsonde data are published and described by Konow et al. (2018a, b, c).

*Author contributions.* FA and SC were initiators of the DFG HAMP project. MG and MW derived the lidar cloud mask and water vapor profiles. HK provided quality controlled HAMP and dropsonde data in a unified file format. MJ and SC conceptualized this study. MM designed the PAMTRA simulations. MJ developed the HAMP retrievals, conducted the analysis, and wrote the paper with support and input from all co-authors.

*Competing interests.* The authors declare that they have no conflict of interest.

*Acknowledgements.* The work has been supported by the German Research Foundation (Deutsche Forschungsgemeinschaft, DFG) within the DFG Priority Program (SPP 1294) "Atmospheric and Earth System Research with the Research Aircraft HALO (High Altitude and Long Range Research Aircraft)" under grant CR111/10-11. We would like to thank Daniel Klocke for running the ICON simulations and the German Climate Computing Centre (DKRZ) for storing and supplying the data. D. Klocke is supported by the Hans Ertel Centre for Weather Research (HErZ) by the DWD. Sabrina Schnitt and Mareike Burba are thanked for comments improving the manuscript. We appreciate the
dedication of Bjorn Stevens as the driving force behind the tropical NARVAL activities.





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
