# Peer review of "Investigating the liquid water path over the tropical Atlantic with synergistic airborne measurements"

_Atmospheric Measurement Techniques, 2019_

## Referee Comment (RC1) · Anonymous Referee #1 · 15 Mar 2019

Marek Jacob et al. present retrievals of integrated water vapour (IWV), liquid water path (LWP) and rain water path (RWP) from airborne passive and active microwave observations collected using the German HALO aircraft during two campaigns over the tropical North Atlantic. The retrievals are developed using artificial neural networks trained on a dataset generated using radiative transfer simulations and atmospheric profiles taken from a cloud-resolving model. The IWV retrievals are evaluated using independent measurements of water vapour from dropsondes and Lidar, and a theoretical comparison using a separate dataset of modelled profiles and brightness temperatures. Analysis of the retrievals from flights conducted during the wet and dry seasons shows higher IWV during the wet season, but a higher frequency of clouds

with larger LWP and RWP during the dry season flights.

A slightly clearer distinction between total liquid water path, cloud liquid water path and rain liquid water path could be made throughout the paper. I would also like to see some more details of the distinction between cloud and rain liquid water path. This is hinted at on P7 line 17, which implies that it is taken from the ICON microphysics scheme, but it would be helpful to specify the difference in terms of the different size distributions etc.

Why are all the available microwave channels not used in the retrieval? I would expect that particularly the 183GHz channels would contain additional information on the IWV, including its vertical distribution, and the quasi-window channels on the far wings of the 118 and 50-60GHz O2 bands will also respond strongly to liquid water. Since the data are already screened for cloud ice then scattering at 183GHz should not be a concern here.

I would like to see some further discussion on the impact of surface wind speed (and the minor impact of surface temperature) on the retrieval of LWP and IWV. How does the frequency-variation of the brightness temperatures differ for surface wind speeds compared to that for IWV/LWP/RWP shown in figure 2? Is there any independent information content on the wind speed contained in the radiometer measurements, or does it effectively just add noise to the LWP/IWV retrievals?

I find it slightly surprising that there is more liquid water during the dry season than the wet. I would like to see some more discussion about how the results in sec. 6 may be influenced by the choice of flight paths during the two campaigns. If specific conditions were either targeted or avoided then this could significantly bias the results.
**Minor points**

The authors note that the WALES IWV measurements are only available in clear sky conditions so they only provide validation for the MWR retrievals when there are no confounding effects from liquid water. Is it possible to split the dropsondes into clear and cloudy scenes to demonstrate if there is any impact of liquid water on the quality of the IWV retrievals?

P6 line 18 Why is there a need to convert from water vapour number density to volume mixing ratio? It is the former that is required to calculate integrated water vapour mass.

P8 figure 2 I suggest using a logarithmic colour scale to show the relative frequency to highlight any detail away from the strong "clear sky" line

P9 line 31 Biases with respect to what?

It would be useful to have an indication of along-track distance on figure 4 rather than just time

P13 final paragraph – it would be nice to refer to fig 6 early in this discussion.

P14 discusses the impact of negative LWP retrieval values on the bias. These could be avoided by performing the retrieval in logarithmic space (I.e. retrieving log(LWP)). Would this have a significant impact on the results?

In figure 10 it might be clearer to plot the LWP and RWP on a logarithmic scale – in the current plot it is hard to see the cloud LWP retrieved by the MWR between 17:38:30 and 17:39:10 that is discussed in the text at the end of page 17

The paper is clearly written and generally easy to follow, although I find a number of sentences do not read well and should be redrafted. There are also a few typos:

P1 line 1 "...identified especially marine low level clouds to play a critical role for the climate."
P1 line 5 "...to better understand the LWP of warm clouds..."

P2 line 5 "Especially, shallow marine clouds are attributed to contribute largely to intermodel

spread of climate models"

P2 line 28 "Visible/near infrared techniques such as those applied to MODIS..."

P3 line 9 "...allow to study clouds with similar, however, more sensitive and higher spatially resolving instrumentation than available on satellites." Perhaps "...allow the study of clouds with similar, but more sensitive and higher spatially resolving, instruments to these available on satellites."

P3 line 13 "Their study shows *the* sub-footprint variability of spaceborne Special Sensor Microwave Imager/Sounder..."

P3 line 24 "The assessment of LWP (Sec. 4) reveals the importance *of using* ancillary measurements, e.g. lidar measurements for low LWP values and cloud radar measurements for lightly precipitating cases."

P3 line 26 "... between dry and wet seasons"

P4 figure 1 I think the caption mis-labels the thin and thick lines based on the dates in the legend (I.e. NARVAL 1 looks like it should be the thick lines)

P5 line 12 "more dominant in the higher *frequency* window channels"

P5 line 16 remove the comma after "both"

P6 line 9 "cloudy conditions as well as possible"

P6 line 25 "...prevent having data during some flights..."

P7 line 10 is ambiguous. Are all profiles with ice excluded, or only ones with ice water path above 1000g/m2?
P7 line 11 "...the ocean are used"

P8 line 3 "... are visible as a line..."

P9 line 11 "...never deviates more than ...."

P14 line 7 "... decided to use a retrieval"

P21 line 11 "... consistent with"

---

## Referee Comment (RC2) · Anonymous Referee #2 · 8 Apr 2019

This manuscript provides a nice overview of the LWP retrievals for the 2 NARVAL campaigns. The documentation will be a useful resource for future publications. I think a bit more discussion of the science possible with the integration of the radar and Lidar datasets, and of the radar data itself would enhance the scientific impact of the manuscript, but these constitute minor comments, incorporated into those provided below. Also, while I am not sure of the Copernicus standards, I would recommend that DOIs be generated for the datasets and included within the manuscript.

Minor comments:

p.2 line 14: also mention the clear-sky contribution to the field of view (it is mentioned

later but the sentence suggest precip is the major error source).

p. 2 Line 27: what do Greenwald et al and other conclude for the tropical Atlantic region you are interested in?

P. 3. some where I think the adiabatic constraint on LWP is worth mentioning. Is it possible to construct an adiabatic estimate from the Lidar cloud top height and drop-sonde RH-derived cloud base do you think? This is an earnest question - I am not sure how well this would work. But it would provide an additional constraint on the retrieval that might be more physical than the imposed 1000 g/mˆ2 (and its relaxation), and could also provide some additional physical insights. For example, in clean marine stratocumulus regions, the adiabatic constraint on LWP seems to hold well until about 200 g/mˆ2, at which point precipitation begins to deplete LWP (Zuidema et all, 2005, fig. 8 and 9). I think during RICO the adiabaticity deviated more quickly from the theoretical maximum because of mixing with environment air (Rauber et al., 2007). Related to this I do not see any discussion on the radar or cloud top height at which precipitation becomes discernible later on in the manuscript - perhaps I missed it.

P. 3 lines 11-19: what is approximately the spatial footprint of the HAMP instrumentation? It would be nice to see this number in relation to the satellite spatial footprints. On p. 6 you mention that the different footprints and sensitivities of the instruments are covered in Stevens et al 2019, but a brief summary here would be useful.

P. 7 line 23: how is scattering off of the ocean surface dealt with?

p.9 line 24: where is the ocean emissivity represented? It would be nice to see a bit more discussion of the ocean surface microwave radiation characteristics in general. A figure of the emission/scattering as a function of SST and wind speed, for the 2 frequencies would be nice, for example. How much does error in the surface characterization contribute to the overall error?

p.13 line 4; Is there any cloud fraction within a model column? At a grid spacing of 0.5

degree, clouds will not necessarily fill the full grid box.

p. 17 : it would be nice to see the retrieved LWP/RWP as a function of the vertically-integrated reflectivity from both campaigns as part of fig. 11. They should look the same, if not, that may tell you something about the cloud droplet number concentration variation between the two seasons.

How does RWP spatial heterogeneity affect the retrieval do you think?

P. 18: how does WVP vary in this example?

p. 19, lines 17-18: Some discussion of the sampling of the diurnal cycle (I presume HALO only flew during the day, were cumuli more prevalent in the afternoon?), and how that might alias into the results from the 2 seasons would be nice. I presume the BCO LWP measurements mentioned are diurnal averages

P.11: an easy additional plot would be how LWP and RWP vary with lidar-derived cloud top height. This would be of scientific interest. How would that compare to, e.g., Byers and Hall, 1955?

P. 23, data availability: do the datasets have dois? They should.

The writing overall is fine, but there are small awkward uses of the English language sprinkled throughout that reflect English as a second language. If it is possible to find a native English speaker to read it that would polish the manuscript. In particular the abstract and its first sentence needs a revisit (you could consider just removing the first sentence).

Other comments on the abstract: mention the frequencies you use. You don't mention the linear regression approach, is that intentional? Mention clear-sky frequency and LWP statistics, as opposed to focusing on IWV - the title only mentions LWP after all. Overall the abstract seems to have been written in a hurry.

References:

[Figure]

Byers, H. R., and R. K. Hall, 1955: A census of cumulus- cloud height versus precipitation in the vicinity of Puerto Rico during the winter and spring of 1953–1954. J. Meteor., 12, 176–178.

Rauber et al., 2007: Rain in Shallow Cumulus over the Ocean, Bull. Am. Meteor. Soc., 88, pp. 1912-1928. doi:10.1175/BAMS-88-12-1912

Zuidema, P., E. Westwater, C. Fairall and D. Hazen, 2005: Ship-based Liquid Water Path Estimates in Marine Stratocumulus. J. Geophys. Res., 110, D20206, doi:10.1029/2005JD005833

---

## Author Comment (AC1) · 21 May 2019

**Authors' Reply**

The authors would like to thank the referees for their constructive feedback, that helped in improving the manuscript. In the following, all revision points are addressed and the resulting text edits are included in the following way:
The comments are repeated and the responses are given below. Changes made in the manuscript are indicated in blue. Figure
5  numbers with "R" correspond to figures in this reply not included in the manuscript.

**Reply to Anonymous Referee #1**

**Comment:** A slightly clearer distinction between total liquid water path, cloud liquid water path and rain liquid water path could be made throughout the paper. I would also like to see some more details of the distinction between cloud and rain liquid water path. This is hinted at on P7 line 17, which implies that it is taken from the ICON microphysics scheme, but it would be
10  helpful to specify the difference in terms of the different size distributions etc.
**Response:** To clarify the discussion we added the following to the the second paragraph of Sec. 1: "Thus, we define LWP as the sum of CLWP and RWP." We checked throughout the manuscript the consistency and corrected especially Sec. 1 accordingly Furthermore, it is made clear that the size distributions from ICON microphysics. They are specified in Sec. 2.3, paragraph 2, now. "Cloud and rain particles are simulated with a $20\,\mu m$ diameter mono-disperse and exponential distribution of water
15  spheres, respectively. The exponential distribution has its intersect $N_0$ classically fixed to $0.08\,cm^{-4}$ (Marshall and Palmer, 1948)."

    **Comment:** Why are all the available microwave channels not used in the retrieval? I would expect that particularly the 183 GHz channels would contain additional information on the IWV, including its vertical distribution, and the quasi-window
20  channels on the far wings of the 118 and 50-60GHz O2 bands will also respond strongly to liquid water. Since the data are already screened for cloud ice then scattering at 183GHz should not be a concern here.
**Response:** We thank the reviewer for proposing this retrieval extension. It certainly is an interesting objective for a follow-up study, but we chose the frequencies for two main reasons. First, we wanted to use only frequency bands that are currently also used by spaceborne microwave sensors to obtain a comparable product. Second, including additional channels to the LWP
25  retrieval inhibits the calibration crosscheck with the IWV retrieval. This is important to avoid additional bias errors, as the calibration and the absorption characteristics in the O2 band are still uncertain (Maschwitz et al., 2013). Furthermore, the 183 GHz channels unfortunately suffered from hardware instabilities during both campaigns as pointed out by Konow et al. (2018a) and their data is not available all the time.

30     **Comment:** I would like to see some further discussion on the impact of surface wind speed (and the minor impact of surface temperature) on the retrieval of LWP and IWV. How does the frequency-variation of the brightness temperatures differ for surface wind speeds compared to that for IWV/LWP/RWP shown in figure 2? Is there any independent information content on the wind speed contained in the radiometer measurements, or does it effectively just add noise to the LWP/IWV retrievals?
**Response:** The error of the LWP retrieval shows no dependence on the 10 m wind speed as it can be seen in Fig. R1. A slight
35  mean underestimation of the retrieval between 5 and $10\,g\,m^{-2}$ for calm wind decreases toward wind speed of about $6\,m\,s^{-1}$. There seems to be a retrieval overestimation for stronger wind speed above $12\,m\,s^{-1}$, but this effect is of minor significance, as wind speed above $12\,m\,s^{-1}$ occurs only rarely (as determined by ICON).
The IWV retrieval error as a function of wind speed is shown in Fig. R2. A slight systematic dependence of the error on wind speed is notable. However, the largest errors are again related to rare wind speeds.
40  To address this issue included a variation of the surface wind speed in Fig. 2. It shows a brightness temperature change of up to 2 to 3 K per $5\,m\,s^{-1}$ wind speed change. The potential to retrieve surface wind speed from this rather weak signal, however, is low. The surface temperature influence of less than 1 K difference for sea surface temperature (sst) variation of 4 K is too weak for any reasonable sst retrieval with the HAMP channels. We added to the 3rd paragraph of Sec. in the manuscript: "The near surface wind speed slightly alters the BTs through modification of surface reflectivity and emissivity as also shown in Fig. 2.
45  This influence will act as a random source of error to the LWP and IWV retrievals as no independent information to correct for

[Figure]

**Figure R1.** Neural network LWP retrieval error as function of 10 m wind speed.

[Figure]

**Figure R2.** Left: Neural network IWV retrieval error as function of 10 m wind speed. Right: 10 m wind speed frequency distribution of the test database.

**Comment:** I find it slightly surprising that there is more liquid water during the dry season than the wet. I would like to see some more discussion about how the results in sec. 6 may be influenced by the choice of flight paths during the two campaigns. If specific conditions were either targeted or avoided then this could significantly bias the results.

**Response:** Indeed we find the higher LWP during the dry season to be one of our most interesting results. While you are right that the flight patterns could be responsible for part of the difference, also changes in the organization of clouds could cause the differences in cloud fraction and LWP. The fact that the medium LWP range from 100 to 400 gm-2 is less frequent in the wet season could be due to the higher degree of organization causing more clear sky areas and more intense clusters with higher amounts of precipitation. In that sense the latter would be missed by our flight patters as we avoided strongly convective scenes with formation of large ice particles. The aspect of organization is currently investigated by different LES modeling groups and will be a major objective of the EUREC⁴A campaign.

The discussion was added to the paper as follows: "The higher LWP in the dry season might partly be explained by the choice

of flight patterns. However, an analysis of ground-based LWP measurements at the Barbados Cloud Observatory (Stevens et al., 2016) over the years 2013-2018 confirms the generally higher LWP values during December than August (not shown). Thus, also changes in the organization of clouds could cause the differences in cloud fraction and LWP. The fact that the medium LWP range from $100\,\mathrm{g\,m^{-2}}$ to $400\,\mathrm{g\,m^{-2}}$ is less frequent in the wet season could be due to the higher degree of organization causing more clear sky areas and more intense clusters with higher amounts of precipitation. In that sense the latter would be missed by our flight patters as we avoided strongly convective scenes with formation of large ice particles."

The abstract was extended by "We hypothesize that higher degree of cloud organization on larger scales in the wet season reduces the overall cloud cover and observed LWP."

Within the conclusions we added: "An extension of the NARVAL observations is planned by the EUREC$^4$A field study in early 2020 ('elucidate the couplings between clouds, convection and circulation'; Bony et al. (2017)) which among other objectives will investigate convective aggregation. The algorithms presented here will be applied and together with additional measurements a better understanding of the governing processes that cause differences between dry and wet season will be analyzed. For that the campaign will provide additional observations of large scale dynamics, horizontally resolved remote sensing observations by a second and in situ observations by additional aircraft in the cumulus layer. Also, more locally targeted flights, distributed over the daytime are planed to study the diurnal cycle. Together with ship, drone and buoy measurements a unique dataset for a better understanding of precipitation onset will be generated."

**Comment:** The authors note that the WALES IWV measurements are only available in clear sky conditions so they only provide validation for the MWR retrievals when there are no confounding effects from liquid water. Is it possible to split the dropsondes into clear and cloudy scenes to demonstrate if there is any impact of liquid water on the quality of the IWV retrievals?

**Response:** The separation between clear sky and cloudy sondes is the reason, why we separate Tab. 2 in "observed pairwise" (all sky) and "observed by all" (WALES available, clear sky). We added the following paragraph to make this more explicit. "A small confounding effect from liquid water in cloudy scenes can be derived from the separation of the HAMP – dropsonde comparison into all ('observed pairwise') and clear sky ('observed by all', i.e., when also WALES is available) in Tab. 2. In the NARVAL1 dataset, the bias for cloudy sky sondes ($0.24\,\mathrm{kg\,m^{-2}}$) is somewhat smaller than that for clear sky ($0.32\,\mathrm{kg\,m^{-2}}$). However, RMSD and STD in cloudy scenes are about $0.3\,\mathrm{kg\,m^{-2}}$ larger than in clear sky. NARVAL2 also shows a larger bias in cloudy sky of about $0.53\,\mathrm{kg\,m^{-2}}$ in comparison to clear sky ($0.28\,\mathrm{kg\,m^{-2}}$). The cloudy sky RMSD and STD of $1.32\,\mathrm{kg\,m^{-2}}$ and $1.21\,\mathrm{kg\,m^{-2}}$, respectively, are only slightly larger than their clear sky counterparts. An increase of the random error for cloudy scenes is expected as also higher water vapor variations are expected in heterogeneous cloud fields."

**Comment:** P6 line 18 Why is there a need to convert from water vapour number density to volume mixing ratio? It is the former that is required to calculate integrated water vapour mass.

**Response:** Indeed, you are right. We note this conversion from water vapor density to water vapor mixing ratio because we want to show mixing ratio in Fig. 4.

**Comment:** P8 figure 2 I suggest using a logarithmic colour scale to show the relative frequency to highlight any detail away from the strong "clear sky" line

**Response:** We followed the suggestion and changed the color scale of Fig. 3, that was on page 8 of the previous manuscript version and shows BT frequency distribution.

**Comment:** P9 line 31 Biases with respect to what?

**Response:** The biases are related to the deviation of LWP from 0 during clear sky scenes. We adjusted the sentence accordingly. "When retrieval algorithms are applied to HALO measurements, slight biases of LWP from 0 with slow changes over time are observed during clear sky scenes."

**Comment:** It would be useful to have an indication of along-track distance on figure 4 rather than just time

**Response:** We followed the suggestion and updated Fig. 4 as well as Fig. 10 accordingly.

**Comment:** P13 final paragraph – it would be nice to refer to fig 6 early in this discussion.
**Response:** We added an earlier reference.

**Comment:** P14 discusses the impact of negative LWP retrieval values on the bias. These could be avoided by performing the retrieval in logarithmic space (I.e. retrieving log(LWP)). Would this have a significant impact on the results?
**Response:** Retrieving log(LWP) would avoid negative values, but it would inevitably result in a overestimation of the retrieved mean LWP as noise is always positive. Another reason for not retrieving the logarithm is, that the natural first order approximation relation between brightness temperatures and LWP is linear.

**Comment:** In figure 10 it might be clearer to plot the LWP and RWP on a logarithmic scale – in the current plot it is hard to see the cloud LWP retrieved by the MWR between 17:38:30 and 17:39:10 that is discussed in the text at the end of page 17
**Response:** We changed Fig. 10 by using a piecewise linear scale now, that magnifies the range from -20 to $20\,\mathrm{g\,m^{-2}}$. We decided against a logarithmic scale as this can not display negative retrieval artifacts.

**Comment:** The paper is clearly written and generally easy to follow, although I find a number of sentences do not read well and should be redrafted. There are also a few typos: [...]
**Response:** All remaining minor corrections by the referee were agreed on and are incorporated in the revised manuscript.

**Comment:** P1 line 1 "...identified especially marine low level clouds to play a critical role for the climate."
**Response:** We removed the first sentence on recommendation from referee #2.

**Comment:** P1 line 5 ". . .to better understand *the* LWP of warm clouds..."
**Response:** "the" is added.

**Comment:** P2 line 5 "Especially, shallow marine clouds are attributed to contribute largely to intermodel spread of climate models"
**Response:** We redrafted this sentence to: "Sherwood et al. (2014) attribute especially shallow marine clouds to contribute largely to intermodel spread of climate models."

**Comment:** P2 line 28 "Visible/near infrared techniques *such* as those applied to MODIS..."
**Response:** "such" is added.

**Comment:** P3 line 9 "...allow to study clouds with similar, however, more sensitive and higher spatially resolving instrumentation than available on satellites." Perhaps "...allow the study of clouds with similar, but more sensitive and higher spatially resolving, instruments to these available on satellites."
**Response:** We thankfully incorporated this comment.

**Comment:** P3 line 13 "Their study shows *the* sub-footprint variability of spaceborne Special Sensor Microwave Imager/Sounder..."
**Response:** We added "the".

**Comment:** P3 line 24 "The assessment of LWP (Sec. 4) reveals the importance *of using* ancillary measurements, e.g. lidar measurements for low LWP values and cloud radar measurements for lightly precipitating cases."
**Response:** We thankfully incorporated this comment.

**Comment:** P3 line 26 ". . . between dry and wet *seasons*"
**Response:** Changed "season" to "seasons".

**Comment:** P4 figure 1 I think the caption mis-labels the thin and thick lines based on the dates in the legend (I.e. NARVAL 1 looks like it should be the thick lines)
**Response:** Corrected.

5   **Comment:** P5 line 12 "more dominant in the higher *frequency* window channels"
**Response:** Corrected "window" to "frequency".

**Comment:** P5 line 16 remove the comma after "both"
**Response:** Removed comma.

10

**Comment:** P6 line 9 "cloudy conditions as *well* as possible"
**Response:** Corrected "good" to "well".

**Comment:** P6 line 25 "...prevent having data during some flights..."
15  **Response:** We redrafted that sentence to: "While all research flights during NARVAL1 can be used, no data is available for some NARVAL2 flight days due to hardware issues as summarized in Tab. 1."

**Comment:** P7 line 10 is ambiguous. Are all profiles with ice excluded, or only ones with ice water path above 1000 g/m$^2$ ?
**Response:** We reordered that sentence for better understanding. "In general, the training and test data excludes cases with LWP
20  greater than $1000\,\mathrm{g\,m^{-2}}$ and cases with ice, i.e., 86 % of all profiles over the ocean are used."

**Comment:** P7 line 11 "...the ocean *are* used"
**Response:** Corrected "is" to "are".

25  **Comment:** P8 line 3 "...are *visible* as a line..."
**Response:** Corrected "visibly" to "visible".

**Comment:** P9 line 11 "...never deviates *more* than..."
**Response:** Added the missing "more".

30

**Comment:** P14 line 7 "...decided *to use* a retrieval"
**Response:** Replaced "we decided for" by "we chose". Further we slightly changed the beginning of the following sentence to:
"Instead to include clear sky directly in the retrieval, we make use of lidar measurements, which are better suited than MWR for cloud masking."

35

**Comment:** P21 line 11 "...consistent *with*"
**Response:** Replaced "to" by "with".

---

## Author Comment (AC2) · 21 May 2019

**Authors' Reply**

The authors would like to thank the referees for their constructive feedback, that helped in improving the manuscript. In the following, all revision points are addressed and the resulting text edits are included in the following way:
The comments are repeated and the responses are given below. Changes made in the manuscript are indicated in blue. Figure
5  numbers with "R" correspond to figures in this reply not included in the manuscript.

**Reply to Anonymous Referee #2**

**Comment:** I think a bit more discussion of the science possible with the integration of the radar and Lidar datasets, and of the radar data itself would enhance the scientific impact of the manuscript, but these constitute minor comments, incorporated into those provided below.

10  **Response:** We added an outlook to combine the results of the present study with radar and lidar datasets to study the presence and condensate loads of different shallow trade wind cumulus types. We added to Sec. 7: "With respect to trade wind cumuli, the products of the present study in combination with cloud boundary estimations from the radar and backscatter lidar will be used to evaluate the condensate loads of different shallow trade wind cumulus types in large eddy simulations. For example, radar and lidar both detect shallow convection or shallow outflow anvils as depicted in Fig. 10. In addition, the lidar also allows

15  detecting boundary layer driven clouds, which have tops around 1 km and are below the radar sensitivity."

**Comment:** Also, while I am not sure of the Copernicus standards, I would recommend that DOIs be generated for the datasets and included within the manuscript.

**Response:** The DOI assignment was in preparation and is completed now. DOIs are added to the references.

20

**Comment:** p.2 line 14: also mention the clear-sky contribution to the field of view (it is mentioned later but the sentence suggest precip is the major error source).

**Response:** We added two sentences to that paragraph to address this point. "Furthermore, the observed LWP per se is an average over the sensors field of view, which is affected by cloud and rain inhomogeneity, and clear sky contribution. Therefore,

25  the spatial resolution is a key information to interpret LWP statistics."

**Comment:** p. 2 Line 27: what do Greenwald et al and other conclude for the tropical Atlantic region you are interested in?

**Response:** It is difficult to give any quantitative estimate from the coarse figures provided by Greenwald et al. (2018) for the North Atlantic tropical region. However, our study region in the tropics behaves close to the global average conditions. Never-

30  theless, from Elsaesser et al. (2017) we added two values. "Elsaesser et al. (2017) additionally estimate the contribution RWP to the total LWP by a simple parametrization and recommend to only use those values with a ratio RWP:LWP of less than 0.2. The average MAC RWP:LWP ratio in our area of interest is 0.23 and 0.30 in December 2013 and August 2016, respectively. Therefore, a more detailed assessment of the rain cloud partitioning is important to better interpret satellite measurements in our study area."

35

**Comment:** P. 3. some where I think the adiabatic constraint on LWP is worth mentioning. Is it possible to construct an adiabatic estimate from the Lidar cloud top height and dropsonde RH-derived cloud base do you think? This is an earnest question - I am not sure how well this would work. But it would provide an additional constraint on the retrieval that might be more physical than the imposed 1000 g/m^2 (and its relaxation), and could also provide some additional physical insights. For

40  example, in clean marine stratocumulus regions, the adiabatic constraint on LWP seems to hold well until about 200 g/m^2, at which point precipitation begins to deplete LWP (Zuidema et all, 2005, fig. 8 and 9). I think during RICO the adiabaticity deviated more quickly from the theoretical maximum because of mixing with environment air (Rauber et al., 2007). Related to this I do not see any discussion on the radar or cloud top height at which precipitation becomes discernible later on in the manuscript - perhaps I missed it.

45  **Response:** Thanks for these interesting thoughts. However, two aspects limit the applicability of the adiabatic theory in our

case. First, the adiabatic assumption requires, that the cloud develops through vertical transport, i.e. is buoyancy driven, and without horizontal exchange. This is more realistic in stratocumulus situations as addressed by Zuidema et al. (2005) than in the trade wind cumuli cases as in our study. Quite often we see clouds in the radar, that are not buoyancy driven which is also the case in the example of Fig. 10. The radar echo between 17:40:00 and 17:41:40 looks more like a shallow outflow of the nearby precipitating core. This shallow outflow anvil is not buoyancy driven, as its radar echo shows no link to the lifting condensation level (lcl), which is roughly at 700 m. Second, the vertical cloud extend $\Delta z$ is very important as the adiabatic water content is proportional to $\Delta z^2$, but the exact estimation of $\Delta z$ is difficult. The estimation of the cloud base hight using the lcl derived from near surface dropsonde data has an accuracy of 50 to 100 m. The cloud top height estimation of shallow cumulus adds additional uncertainty as the lidar sees the cloud top typically 250 m higher in altitude than the radar, in cases in which the radar sees a cloud at all. Thus we think, that a comparison to a somehow estimated adiabatic LWP raises more methodological questions than it would help to constrain the LWP.

**Comment:** P. 3 lines 11-19: what is approximately the spatial footprint of the HAMP instrumentation? It would be nice to see this number in relation to the satellite spatial footprints. On p. 6 you mention that the different footprints and sensitivities of the instruments are covered in Stevens et al 2019, but a brief summary here would be useful.
**Response:** The reference to Schnitt et al. (2017) reads now: "Their study uses the 1 km resolution HAMP data to show the sub-footprint variability of spaceborne CLWP estimation of about 30 km resolution. Further they illustrate how MODIS products at 1 km resolution likely underestimate CLWP of thick clouds due to MODIS' sensitivity towards the upper part of the cloud."

**Comment:** P. 7 line 23: how is scattering off of the ocean surface dealt with?
**Response:** We understand the "scattering off of the ocean" as surface reflection. Surface reflectivity is also calculated using FASTEM5. The related sentence in Sec. 2.3 reads now: "The emissivity and reflectivity of the sea ocean surface is calculated by the FAST microwave Emissivity Model version 5 (FASTEM5; Liu et al. (2011)), which is a modification of the Fresnel coefficients including corrections for ocean surface roughness and foam building as a function of wind speed."

**Comment:** p.9 line 24: where is the ocean emissivity represented? It would be nice to see a bit more discussion of the ocean surface microwave radiation characteristics in general. A figure of the emission/scattering as a function of SST and wind speed, for the 2 frequencies would be nice, for example. How much does error in the surface characterization contribute to the overall error?
**Response:** Emission by the surface is only implicitly included in the retrieval. When generating the database, ocean emissivity is calculated by FASTEM5 using sea surface temperature (sst) and 10 m wind speed as input. Based on a comment by referee #1 we investigated the uncertainty due to sst and 10 m wind speed. (See Fig. R1 and R2 and discussion in our answer to referee #1.)

**Comment:** p.13 line 4; Is there any cloud fraction within a model column? At a grid spacing of 0.5 degree, clouds will not necessarily fill the full grid box.
**Response:** This is probably a misunderstanding. The ICON simulations were run at 1.25 km resolution. Afterwards, the data used for the retrieval database was coarse grained to 0.5° as explained in the first paragraph of Sec. 2.3.

**Comment:** p. 17 : it would be nice to see the retrieved LWP/RWP as a function of the vertically integrated reflectivity from both campaigns as part of fig. 11. They should look the same, if not, that may tell you something about the cloud droplet number concentration variation between the two seasons.
**Response:** The RWP retrieval already includes the vertically integrated reflectivity $Z_{int}$. Therefore, the empirical relation between RWP and $Z_{int}$ would be established from two dependent variables. Each of our retrievals was trained for each campaign individually by using simulations for the respective period. Thus, differences in the RWP-$Z_{int}$ relation would also represent the different training datasets.
Technically, we have to exclude scenes with no radar echo above noise level because $Z_{int} = 0$ can not be represented on a decibel scale from such comparison. A logarithmic scale is required for displaying in analogy to dBZ. This means that such a figure excludes clouds, that were to thin to be detectable by the radar but were detected by the lidar. Nevertheless, we prepared Fig. R1 showing the relation between LWP and $Z_{int}$ for scenes, where $10\log_{10}(Z_{int}) > -30$. The scatter plot shows, that there

are less scenes with LWP > 300 g m$^{-2}$ during NARVAL1 than NARVAL2 as it can also be seen on Fig. 11. For NARVAL1, there is a pronounced maximum of combinations for $Z_{int}$ from 15 dB at 100 g m$^{-2}$ increasing to 40 dB at LWP > 400 g m$^{-2}$. A similar relation can be seen also for NARVAL2. In addition there is an second mode for LWP < 200 g m$^{-2}$ with $Z_{int}$ being about 0 dB. Scenes with lower $Z_{int}$ most likely consist of smaller droplets for the same LWP as $Z_{int} \propto D^6 \Delta z$ and LWP $\propto D^3 \Delta z$.

5 So, probably clouds with smaller droplets were slightly more prominent during NARVAL1 than NARVAL2.

Figure R1 is an interesting starting point for a microphysical study. However, to present this topic in an appropriate manner more work has to be done, which will be included on a follow-up study.

[Figure]

**Figure R1.** Decibel of vertically integrated reflectivity ($10 \log_{10}(Z_{int})$) versus LWP during (left) NARVAL1 and (right) NARVAL2.

**Comment:** How does RWP spatial heterogeneity affect the retrieval do you think?

10 **Response:** The spatial heterogeneity of rain affects the airborne HAMP measurements less than microwave satellites as the HAMPs spatial resolution is at least an order of magnitude better than satellite resolution. To illustrate the scale on that HAMP resolves precipitation, we added a km scale to Fig. 10. Also we added to the fourth paragraph of Sec. 5: "The figure shows how HAMP is able to resolve spatial features of showering cells, which were observed with a cross section of several HAMP footprints."

15

**Comment:** P. 18: how does WVP vary in this example?

**Response:** See Fig. R2. We added the summary "the IWV varies around $31.5 \pm 1.5$ kg m$^{-2}$ in this scene" to the Fig. 10 caption in the manuscript, as the main aspect of the figure is the liquid phase and IWV variation is only of secondary interest in that example.

20

[Figure]

**Figure R2.** As Fig. 10 but with additional time series of IWV.

**Comment:** p. 19, lines 17-18: Some discussion of the sampling of the diurnal cycle (I presume HALO only flew during the day, were cumuli more prevalent in the afternoon?), and how that might alias into the results from the 2 seasons would be nice. I presume the BCO LWP measurements mentioned are diurnal averages

**Response:** Yes you are right, we added that "flights were scheduled during local daytime" to the second paragraph of Sec. 2.1. In the conclusions we added that "sound conclusions on the diurnal cycle can not be drawn from the data presented here, as the spatial variability of the clouds on the observed mesoscale was higher than an expected effect of the diurnal cycle." Radar time-height-plots of the NARVAL1 flights that were flown from Barbados to the East and back are rather symmetrical to the return point due to large-scale patterns. This means, a potentially diurnal cycle during the flights is overlaid by the changes in the larger scale cloud field. Additionally, we added in the outlook the following with respect to EUREC4A: "Also, more locally targeted flights, distributed over the daytime are planed to study the diurnal cycle."

**Comment:** P.11: an easy additional plot would be how LWP and RWP vary with lidar-derived cloud top height. This would be of scientific interest. How would that compare to, e.g., Byers and Hall, 1955?

**Response:** Indeed, liquid condensate load versus cloud extend is an interesting comparison. Figure R3 depicts the relations during the two campaigns. The overall impression of increasing rain amount with increasing cloud top height during NARVAL1 agrees with the findings by Byers and Hall (1955). The dry winter season during the NARVAL1 campaign compares best to their pioneer study according to their description. Differences exist in details and but are also partially due to the analysis approach. Byers and Hall subjectively identified cloud objects while we analyze the data profile-wise. For example, Byers and Hall (1955) found the lowest cloud top of a precipitating cloud near 1.8 km, whereas we already observed RWP > 10 g m$^{-2}$ for cloud top heights near 1.0 km. Stevens et al. (2019, in press) present in Fig. 7 a cloud object oriented analysis of NARVAL, that directly uses radar reflectivity instead of the RWP retrieval presented in our manuscript. A more detailed analysis of cloud dimensions in relation to their LWP and RWP will follow in a subsequent study, which is in preparation at the moment.

[Figure]

**Figure R3.** Lidar (WALES) derived cloud top height in relation to LWP (left) and RWP (right) for both campaigns.

**Comment:** P. 23, data availability: do the datasets have dois? They should.

**Response:** The DOI assignment was in preparation and is completed now. DOIs are added in the references.

**Comment:** The writing overall is fine, but there are small awkward uses of the English language sprinkled throughout that reflect English as a second language. If it is possible to find a native English speaker to read it that would polish the manuscript.

**Response:** Referee #1 pointed out problematic sentences and typos that are remedied now.

**Comment:** In particular the abstract and its first sentence needs a revisit (you could consider just removing the first sentence).

Other comments on the abstract: mention the frequencies you use. You don't mention the linear regression approach, is that intentional? Mention clear-sky frequency and LWP statistics, as opposed to focusing on IWV - the title only mentions LWP after all. Overall the abstract seems to have been written in a hurry.

**Response:** We removed the first sentence from the abstract as suggested and rearranged most parts of the abstract for improved comprehensibility. We incorporated the radiometer frequencies into the abstract. We don't mention the linear regression approach on purpose, as it is mainly used as reference to classical retrievals. Averages for LWP, RWP and cloudiness were added and the sentence order was rearranged such that it has the dry and wet season in a consistent order. Further, we reformulated the closing two sentences of the abstract after the discussion of the flight patterns question by referee #1 and reconsidering that issue. We conclude that our former formulation was to negative. The revised abstract reads now:

"Liquid water path (LWP) is an important quantity to characterize clouds. Passive microwave satellite sensors provide the most direct estimate on global scale, but suffer from high uncertainties due to large footprints and the superposition of cloud and precipitation signals. Here, we use high spatial resolution airborne microwave radiometer (MWR) measurements together with cloud radar and lidar observations to better understand the LWP of warm clouds over the tropical North Atlantic. The nadir measurements were taken by the German High Altitude and Long range research aircraft (HALO) in December 2013 (dry season) and August 2016 (wet season) during two Next generation Advanced Remote sensing for VALidation (NARVAL) campaigns.

Microwave retrievals of integrated water vapor (IWV), LWP and rain water path (RWP) are developed using artificial neural network techniques. A retrieval database is created using unique cloud-resolving simulations with 1.25 km grid spacing. The IWV and LWP retrievals share the same eight MWR frequency channels in the range from 22 GHz to 31 GHz and at 90 GHz as their sole input. The RWP retrieval combines active and passive microwave observations and is able to detect drizzle and light precipitation. The comparison of retrieved IWV with coincident dropsondes and water vapor lidar measurements shows root-mean-square deviations below $1.4\,kg\,m^{-2}$ over the range from $20\,kg\,m^{-2}$ to $60\,kg\,m^{-2}$. This comparison raises the confidence in LWP retrievals which can only be assessed theoretically. The theoretical analysis shows that the LWP error is constant with $20\,g\,m^{-2}$ for LWP below $100\,g\,m^{-2}$. While the absolute LWP error increases with increasing LWP, the relative one decreases from 20 % at $100\,g\,m^{-2}$ to 10 % at $500\,g\,m^{-2}$. The identification of clear sky scenes by ancillary measurements, here backscatter lidar, is crucial for thin clouds (LWP $< 12\,g\,m^{-2}$) as the microwave retrieved LWP uncertainty is higher than 100 %.

The analysis of both campaigns reveals that clouds were more frequent (47 % vs. 30 % of the time) in the dry than in the wet season. Their average LWP ($63\,g\,m^{-2}$ vs. $40\,g\,m^{-2}$) and RWP ($6.7\,g\,m^{-2}$ vs. $2.7\,g\,m^{-2}$) were higher as well. Microwave scattering of ice, however, was observed less frequently in the dry season (0.5 % vs. 1.6 % of the time). We hypothesize that higher degree of cloud organization on larger scales in the wet season reduces the overall cloud cover and observed LWP. As to be expected, the observed IWV clearly shows that the dry season is on average less humid than the wet season ($28\,kg\,m^{-2}$ vs. $41\,kg\,m^{-2}$). The results reveal that the observed frequency distributions of IWV are substantially affected by the choice of the flight pattern. This should be kept in mind when using the airborne observations to carefully mediate between long-term ground-based and spaceborne measurements to draw statistically sound conclusions. "